# Deposition and water repelling of temperature-responsive nanopesticides on leaves

Jie Tang[1], Xiaojing Tong[1], Yongjun Chen[2], Yue Wu[1], Zhiyuan Zheng[1], A. Basak Kayitmazer [3], Ayyaz Ahmad[4], Naveed Ramzan[5], Jintao Yang[6], Qingchun Huang[2] & Yisheng Xu [1] ✉

Pesticides are widely used to increase agricultural productivity, however, weak adhesion and deposition lead to low efficient utilization. Herein, we prepare a nanopesticide formulation (tebuconazole nanopesticides) which is leaf-adhesive, and water-dispersed via a rapid nanoparticle precipitation method, flash nanoprecipitation, using temperature-responsive copolymers poly-(2-(dimethylamino)ethylmethylacrylate)-*b*-poly(ε-caprolactone) as the carrier. Compared with commercial suspensions, the encapsulation by the polymer improves the deposition of TEB, and the contact angle on foliage is lowered by 40.0°. Due to the small size and strong van der Waals interactions, the anti-washing efficiency of TEB NPs is increased by 37% in contrast to commercial ones. Finally, the acute toxicity of TEB NPs to zebrafish shows a more than 25-fold reduction as compared to commercial formulation indicating good biocompatibility of the nanopesticides. This work is expected to enhance pesticide droplet deposition and adhesion, maximize the use of pesticides, tackling one of the application challenges of pesticides.

Agriculture is a crucial source of food worldwide, and pesticides are widely used to boost crop yields to meet the global demand for food. However, due to rain wash-off, photolysis, chemical degradation, and surface run-off, only a small fraction (1–25%) of the active ingredient (AI) is used for target organisms[1–3]. The weak adhesion of pesticides droplets to crop foliage is the primary reason for the wastage associated with pesticide spraying[4,5]. In addition, biotic (plant pathogens, inefficient pesticide use) and abiotic (extreme weather such as heatwaves, heavy rains, etc.) stressors are causing global crop yields to steadily decrease by about 50%[3,6]. Therefore, there is an urgent need to improve the adhesion of nanopesticides (NPs) to foliage, which can facilitate the amplifying of agricultural productivity and enhance resistance to climate change[7].

Both pesticide droplet deposition and adhesion to foliage against rainfall erosion contribute to improved pesticide drop retention[8]. Various multifunctional nanocomposite supports or additives, such as carbon nanomaterials[9], silica nanoparticles[10], nanogels[11], polymers[4], have been developed to improve pesticide droplet deposition. For example, graphene oxide was modified with $Cu_{2-x}Se$ nanocrystals for pesticide delivery, which can control the loss of pesticide on the leaf by the piercing effect and anchoring ability of graphene oxide[12]. Zhang et al. reported the preparation of a nano-spinosad by using spiky silica hollow nanoparticles to load spinosad, which maintained the rough surface topology, and demonstrated an attractive dual adhesion property toward both cattle hides and pest surface[13]. A series of temperature-dependent nanogels with tunable structure and high

[1]State Key Laboratory of Chemical Engineering, East China University of Science and Technology, Shanghai 200237, P. R. China. [2]Shanghai Key Lab of Chemical Biology, School of Pharmacy, East China University of Science and Technology, Shanghai 200237, P. R. China. [3]Department of Chemistry, Bogazici University, Istanbul, Turkey. [4]Department of Chemical Engineering, Muhammad Nawaz Sharif University of Engineering and Technology, Multan, Pakistan. [5]Faculty of Chemical, Metallurgical, and Polymer Engineering, University of Engineering & Technology, Lahore, Pakistan. [6]College of Materials Science & Engineering, Zhejiang University of Technology, Hangzhou 310014, P. R. China. ✉e-mail: yshxu@ecust.edu.cn

deformability was successfully developed by using thermosresponsive polymer (PNIPAM) and these nanogels exhibited better foliar wettability with a stable network structure[11]. Meanwhile, the use of nanocarriers such as attapulgite aggregates[14], phosphorylated zein (P-zein)[15], and polydopamine nanoparticles[16] can enhance pesticide adhesion. However, due to complex formulations, low drug-loading efficiency, poor degradation properties, or a lack of strong interactions, the above methods are still ineffective in addressing the main challenges of pesticide overuse. Therefore, it is hypothesized that NPs prepared using materials with great biocompatibility will reduce the contact angle between pesticide and foliage during the pesticide spraying period through electrostatic attraction, hydrogen bonding, and other van der Waals interactions, while making the leaf hydrophobic as a result of a stimuli-response after the pesticide spraying, reducing the effect of rainwater washing on pesticides.

To address the challenges of complex formulations and low drug loading capacity in preparation of NPs, we have employed a technique called flash nanoprecipitation (FNP), a rapid and efficient nanoparticle precipitation method that has shown great promise in producing stable and small dispersed NPs with controllable surface features[17–24]. During the FNP process, a water-miscible organic solvent phase (such as tetrahydrofuran) containing hydrophobic AI and amphiphilic polymers is rapidly mixed with the antisolvent (water) in a confined impinging jet mixer with dilution (CIJ-D), which instantaneously forms a high supersaturation of hydrophobic molecules within milliseconds to form hydrophobic cores. Subsequently, the amphiphilic molecules provide spatial stability, which greatly inhibits further particle growth and agglomerates[22,25].

In this study, we have successfully prepared leaf-adhesive, water-stable, small particle size, and well-dispersed tebuconazole (an ergosterol synthesis inhibitor, TEB) NPs using the FNP method with the block copolymer poly-(2-(dimethylamino)ethylmethylacrylate)-*b*-poly(ε-caprolactone) (PDMAEMA-*b*-PCL) as the carrier, which has charged and temperature-responsive groups[26] allowing the efficient disposition of NPs at low temperatures and with good water repelling properties as the temperature is increased, as shown in Fig. 1. The encapsulation by the polymer improved the dispersion and photostability of TEB, and enhanced the wettability and adhesion ability to leaves due to small size, high specific surface area, and strong

electrostatic interactions. Thermosensitive polymer PDMAEMA has a low critical solubility temperature (LCST) of ~40 °C, which is hydrophilic below the LCST temperature while hydrophobic above the LCST temperature, and this property of PDMAEMA allows us to make the leaf hydrophobic as a result of a stimuli-response[27,28] achieving effective water repelling features. In addition, the slightly acidic apoplastic space in plants allows preferential release of anti-pathogen and protective agent carriers under low-pH conditions which is desirable for pH-dependent PDMAEMA with a p$K_a$ of ~7.4[7]. To mimic the in vivo environment of plants, the fungicidal activity and in vitro responsive release of NPs were also studied. This work is expected to provide a simple and rapid method for the application of TEB, which can enhance pesticide droplet deposition and adhesion, improve pesticide adaptability to extreme weather conditions, and maximize the use of pesticides, tackling one of the application challenges of modern pesticides.

## Results and discussion
### Preparation and characterization of TEB NPs
The size, size distribution, and morphology of NPs have a great influence on the efficiency of pesticide bioactivity. It has been reported that, when the pesticide particles were reduced from around 5 µm to 100 nm by using nanotechnology, the small size effect of nanomaterials can reduce foliar pesticide shedding and improve pesticide utilization, meanwhile, the smaller particle size enlarges the contact area with the leaf which enhances the interaction between NPs and the leaves, therefore improving the adhesion[3].

The size distribution, EE, DLC, and stability of NPs were controlled by varying the copolymer concentration, as shown in Fig. 2a–c, and varying the flow rate, as shown in Fig. 2d–f. The homogeneous dispersion of sufficient PDMAEMA-*b*-PCL around a higher flow rate quickly terminated the growth of TEB and resulted in smaller particle sizes. High EE indicated that almost all TEB was encapsulated by PDMAEMA-*b*-PCL upon precipitation at the millisecond scale which could be attributed to the low solubility ($3.2 \times 10^{-2}$ g/L at 20 °C) of hydrophobic TEB in water. The TEB molecules nucleated and grew at the hydrophobic chain of the polymer during the turbulent mixing process, but at a certain size, the growth was inhibited through steric inhibition inside the

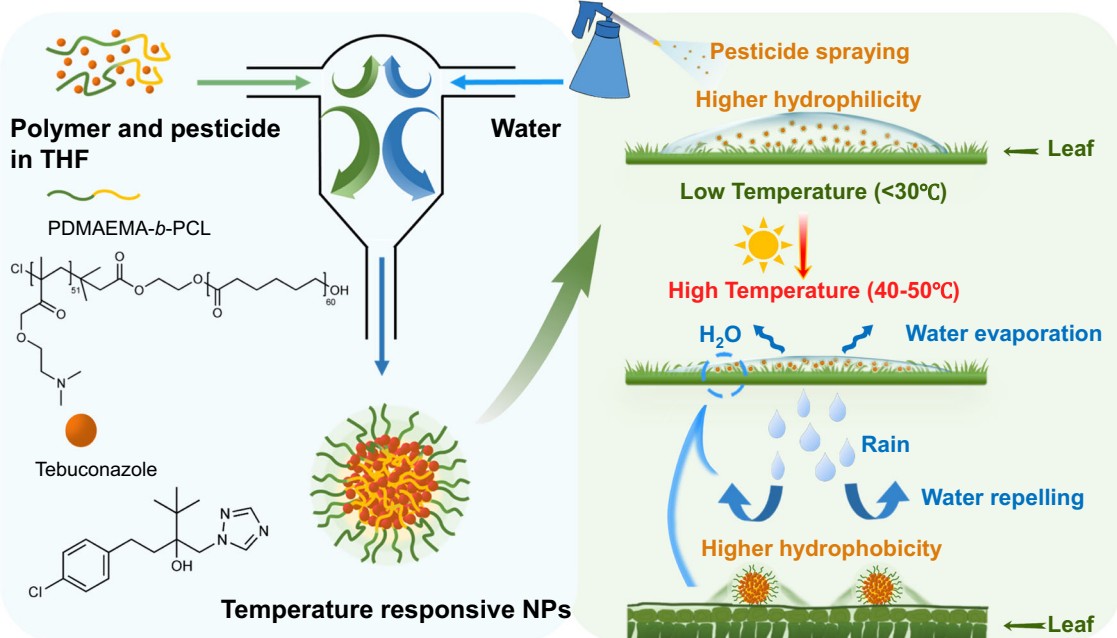

**Fig. 1 | Nanopesticide fabrication for water-repelling on leaves.** Preparation of TEB NPs and the application of nanopesticides on leaves at different temperatures.

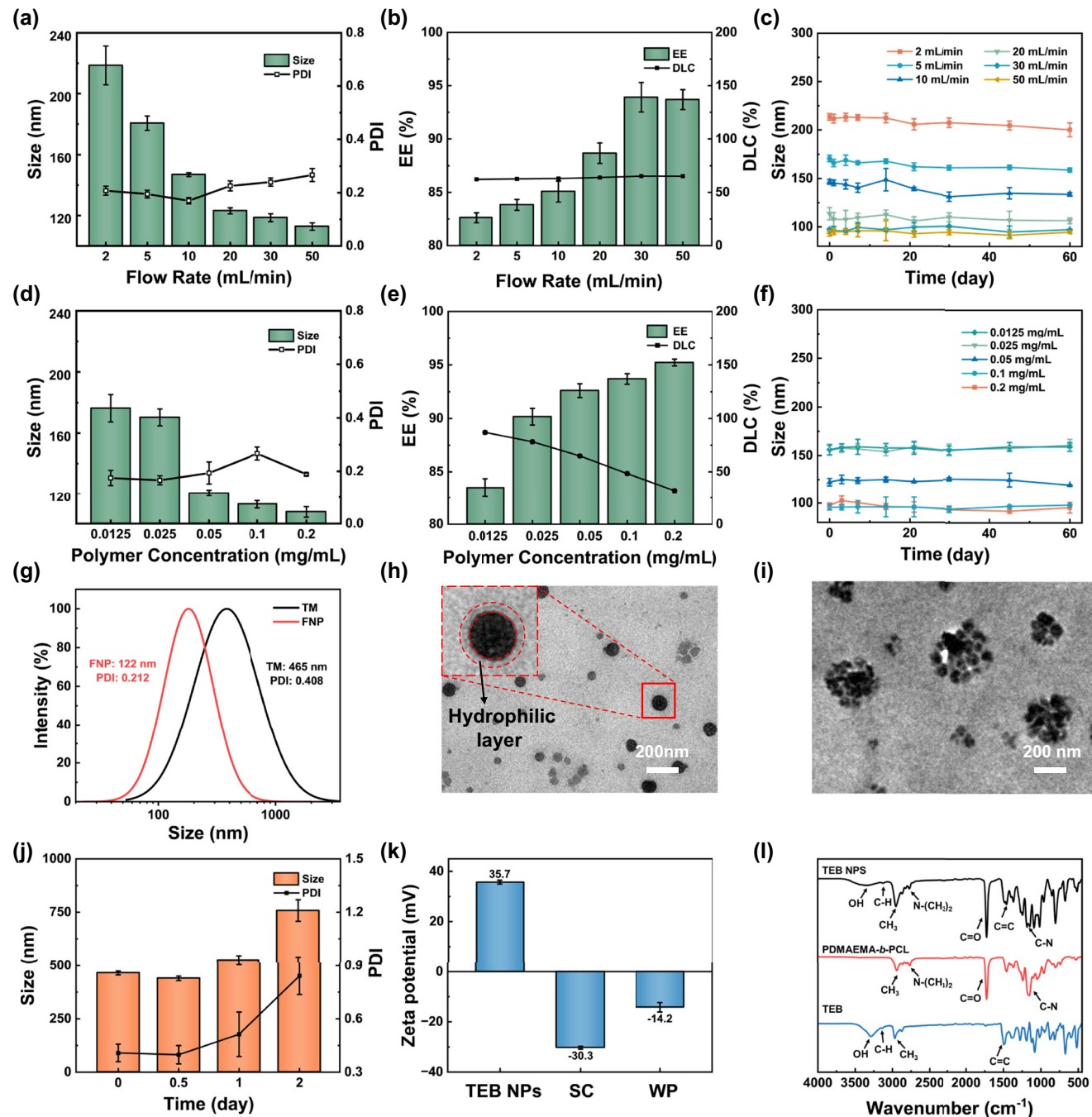

**Fig. 2 | Characterizations of TEB NPs. a** Size distribution, **b** encapsulation efficiency (EE) and drug loading capacity (DLC), and **c** stability of TEB NPs prepared using different flow rates (*n* = 3 independent experiments. Data are presented as mean values ± SD). **d** Size distribution, **e** EE and DLC, and **f** stability of TEB NPs prepared using different polymer concentrations (*n* = 3 independent experiments, Data are presented as mean values ± SD). **g** Size distribution of TEB NPs via FNP vs. TM method. TEM images of TEB NPs via **h** FNP method, and **i** TM method, with 0.1 mg/mL TEB and 0.05 mg/mL PDMAEMA-*b*-PCL (each experiment was repeated three times independently with similar results); **j** stability (*n* = 3 independent experiments, Data are presented as mean values ± SD) of TEB NPs via TM method. **k** Zeta potentials (*n* = 3 independent experiments, Data are presented as mean values ± SD) of TEB NPs, SC (suspension concentrate), and WP (wettable powder). **l** FTIR spectra of TEB NPs, PDMAEMA-*b*-PCL, and TEB.

polymer chains. Hydrophobic TEB, most of which was encapsulated inside the hydrophobic core, was limited to diffuse out from the core of the NP as the organic solvent was dialyzed. With the increase of the amount of PDMAEMA-*b*-PCL and the flow rate, the EE of TEB NPs was increased. The size of NPs produced by FNP was 122 nm with a PDI value of 0.212 indicating a narrow size distribution. After storing for two months, the average size remained almost unchanged, and the limited variation of PDI also indicated the excellent storage stability of TEB NPs in suspension at room temperature

(25 °C). As shown in Supplementary Fig. 1, the average size and PDI of TEB NPs remained essentially constant at 0 °C, and 38 °C during storing for 3 weeks, while the size of TEB NPs at 54 °C was increased from 75 to 200 nm, PDI was increased from 0.16 to 0.3, as the LCST of the polymer was ~40 °C, beyond which the particles gradually aggregated. The superior stability of NPs in suspension could be attributed to the tight binding between the highly hydrophobic TEB core and the hydrophobic chain copolymer as well as the extension of the positively charged PDMAEMA chain in water.

The size distributions of TEB NPs prepared via FNP and TM were shown in Fig. 2g, and NPs prepared by TM displayed larger size and polydispersity. The corresponding TEM image indicated that the morphology of NPs via FNP was spherical as presented in Fig. 2h and Supplementary Fig. 2, with a mean diameter of ~100 nm, while the morphology of NPs via TM was irregular and obvious aggregation was noticed as shown in Fig. 2i. In addition, after two days, the size of NPs via the TM method significantly increased from 410 nm to about 700 nm, as shown in Fig. 2j. In contrast to the conventional thermal dynamic assembling process, the characteristic mixing time during the FNP process can be in the range of milliseconds, which can provide a more uniform supersaturation distribution throughout the whole volume. The much more homogenous supersaturation distribution ensures that the nucleation and growth of NPs are much more uniform, which could decrease the size and size distribution of the as-prepared NPs. After equilibrating EE, DLC, particle size, and stability, the copolymer concentration for subsequent experiments was chosen to be 0.05 mg/mL, and the flow rate was 30 mL/min, which greatly reduced the use of the polymer in the FNP process.

The zeta potentials of different TEB formulations were shown in Fig. 2k, and the TEB NPs had a positive potential of 35.7 mV, and commercial TEB formulations had a negative potential of −30.3 mV and −14.2 mV for SC (suspension concentrate) and WP (wettable powder), respectively. Biological surfaces are often negatively charged, and by using cationic polymers to make NPs positively charged, the electrostatic attraction between the pesticide formulation and the target substrate can be increased to improve the deposition efficiency of the pesticide on the target surface[4].

FTIR was used to identify possible interactions between the fungicides and the NPs, as well as to determine whether the components of the formulations were altered during the preparation process, as shown in Fig. 2l. The spectrum for TEB showed C−H stretching vibration of the benzene ring at 3142 cm$^{-1}$, the C=C vibration of the benzene ring between 1570 and 1501 cm$^{-1}$, and a band at 3291 cm$^{-1}$ related to the hydroxyl group of the TEB molecule. PDMAEMA-$b$-PCL showed characteristic bands including the stretching of ester carbonyls (C=O) at 1725 cm$^{-1}$, C−N at 1150 cm$^{-1}$, and N-(CH$_3$)$_2$ at 2760 cm$^{-1}$, as well as C−H stretching of saturated carbon present in the polymeric chains between 3000 and 2900 cm$^{-1}$. The characteristic peak of TEB appearing at 3219 cm$^{-1}$ and 1510 cm$^{-1}$ in TEB NPs indicated the successful incorporation of TEB into the NPs.

## Stimuli-response properties of TEB NPs

Temperature-responsive nanocarrier systems have great application potential in agriculture as the NPs may release AI to promote crop growth during hot and fast-growing summer months (for instance, 28.8 °C to 50 °C) when the threat of target fungal pathogens and the competition for resources among competing organisms (weeds) are both intense. Since the plant apoplastic pH is normally 5.0–6.0[29], pH-responsive NPs can potentially protect plants from pathogen infection under stress by releasing AI in the apoplastic space of stressed plants. From this perspective, pH/temperature dual-responsive NPs become crucial concerning the thermally responsive deposition properties and controlled release.

The controllable release of pesticide in response to environmental stimuli is highly desirable for improved efficacy and fewer toxic effects. According to the dual-response of the copolymer, the effects of pH and temperature on the TEB NPs release were also studied. At pH 6.0, the effect of temperature on the release profiles of TEB from microcapsules was investigated. Figure 3a showed a trend of increased pesticide release with temperature. The cumulative release obtained at 50 °C is 94.5%, higher than 73.6% at 30 °C and 80.2% at 40 °C after 48 h. Non-covalent interactions between hydrophobic pesticide and polymeric material mainly affect the release profile, i.e., these non-covalent interactions between TEB and its carrier are primarily van der Waals

interactions, which decrease with increasing temperature, facilitating the pesticide release[30]. The reason for the temperature-induced variation in the release rate is that TEB diffusion across the carrier may have a distance-limiting mechanism. The copolymer shell becomes swollen and thick at low temperatures, resulting in a slow-release rate. On the other hand, when the copolymer shell collapses at high temperatures, the thin shell allows for a dramatic increase in the TEB release rate[31].

The release profile of TEB under three different pH values of 4.5, 6.0, and 7.4 at 30 °C is shown in Fig. 3b. At pH 4.5, the cumulative release reached up to 44% after the first 5 h, and continuously increased to about 80% after 48 h; at pH 6.0, the cumulative release attained 40% and 73% after 5 h and 48 h, respectively; at pH 7.4, the cumulative release attained 44% after 48 h. These results showed that the cumulative release of TEB was higher in an acidic medium than in a neutral solution. As shown in Supplementary Fig. 3, the particle size of TEB NPs decreased from 160 nm to 95 nm as the pH values increase from 4.5 to 7.4. The increase in the acidity of the dialysate promoted the protonation of -N(CH$_3$)$_2$ groups in the polymer PDMAEMA-$b$-PCL, which consequently increased intermolecular and intramolecular electrostatic repulsion interactions, and the polymer chains were fully stretched to form channels, thus promoted the release of the pesticides[32–34]. With a high encapsulation efficiency, the TEB molecules were exposed especially at acidic conditions as the hydrophilic chain becomes protonated and the PDMAEMA chains are more expanded in water. This is why the molecules could release faster at the initial stage[34]. The release rate changed significantly with time at different pH values.

The thermal stability of TEB NPs was studied using TGA-DTG (Fig. 3c, d). The mass loss of each substance before 100 °C was caused by the evaporation of adsorbed water and crystal water. For TEB, the rapid decrease in mass fraction at 300–360 °C indicated that TEB was easy to decompose within such a temperature range. PDMAEMA-$b$-PCL presented two stages of decomposition, the first one occurring between 220 °C and 340 °C, attributed to the loss of side groups -CH$_2$CH$_2$N(CH$_3$)$_2$ fragment of the PDMAEMA, and the second stage occurring at 350 °C between 440 °C, attributed to the overlap of two processes, namely thermal decomposition of the side groups and initiation of the disintegration of the main chain. The thermal stability of TEB NPs was studied, which suggested the high stability (up to 300 °C) towards thermal decomposition, and the curve was more similar to the TGA curve of PDMAEMA-$b$-PCL.

Exposing free TEB to UV light resulted in a reduced half-life with decreased pesticidal efficacy. The photostability of TEB was investigated by protecting the photosensitive AI from UV light. To show the improved photostability through NPs and the photodegradation of TEB NPs via the FNP method, free TEB in the same solvent (THF/water = 10:90 $v/v$) with TEB NPs, SC, and WP were investigated under ultraviolet light. The degradation profiles are shown in Fig. 3e, f. Photolysis of TEB could be characterized by a first-order kinetic equation (Supplementary Table 1). In the presence of a UV lamp (E$_{max}$ = 254 nm, 16.4 W/m$^2$), the photolytic rate constant (k, h$^{-1}$) was 0.0498 for TEB NPs, while k was 0.1382 for free TEB, 0.1065 for WP, and 0.0857 for SC, respectively. The half-life times (t$_{1/2}$) of photodegradation for free TEB, WP, SC, and TEB NPs were 5.02, 6.51, 8.09, and 13.92 h, respectively, which indicated the increased photostability by encapsulation via the FNP. Compared to free TEB and commercial pesticides (SC and WP), improved photostability through encapsulation was possible since microcapsules reduce the probability of direct contact between pesticides and incident light. The polymeric material not only blocked out the light but also wrapped and controlled the release of TEB. These results demonstrated that microencapsulation could dramatically improve the photostability of TEB, enhance the utilization efficiency and reduce the input of such agriculture chemicals.

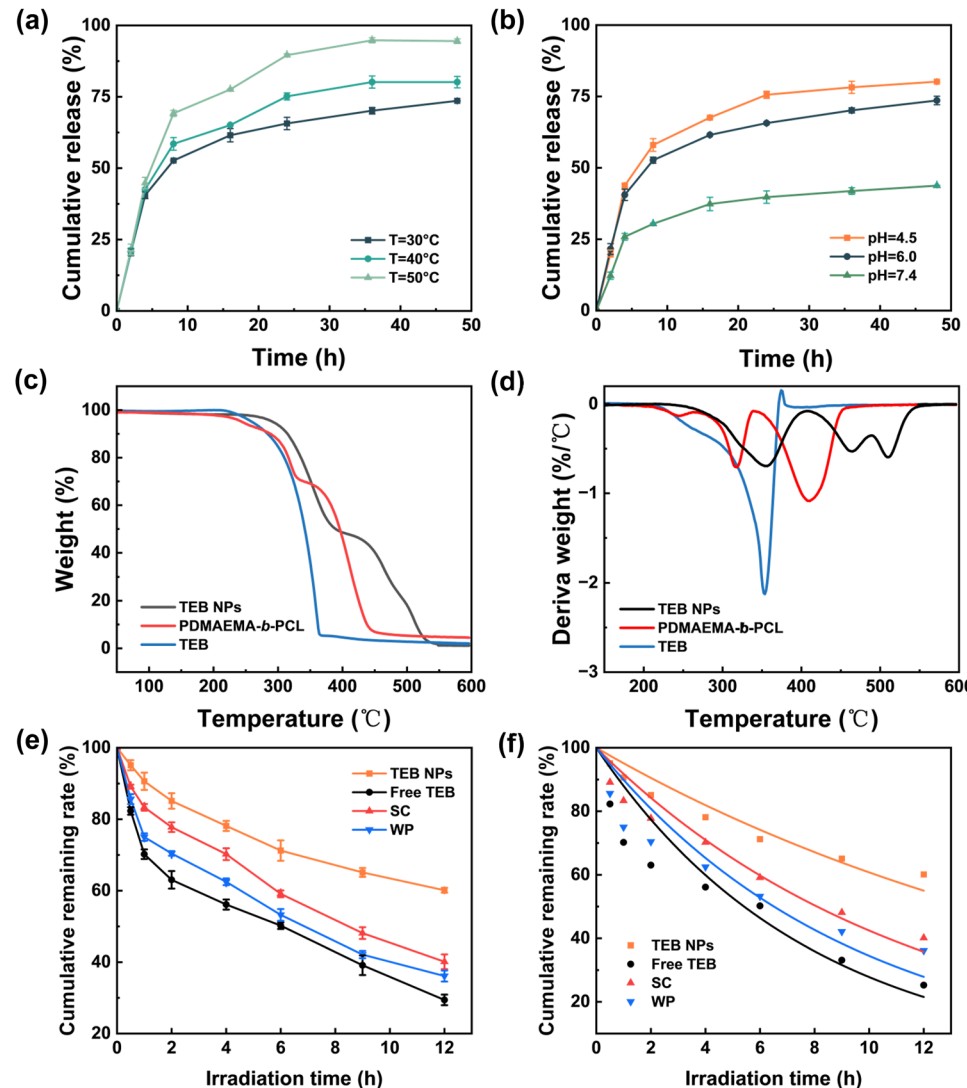

**Fig. 3 | Stimuli-response and photostability properties of TEB NPs.** TEB release at different **a** temperatures at pH 6.0 ($n = 3$ independent experiments, Data are presented as mean values ± SD), and **b** pH values at 30 °C ($n = 3$ independent experiments, Data are presented as mean values ± SD). **c** TGA and **d** DTG thermograms of TEB NPs, PDMAEMA-*b*-PCL, and TEB. **e** Change of TEB cumulative remaining rate with time after UV irradiation ($E_{max} = 254$ nm, $n = 3$ independent experiments, Data are presented as mean values ± SD). **f** Degradation kinetic model of different TEB formulations.

## Wetting and foliar affinity of nanopesticides

SEM was used to characterize the deposition of various TEB formulations on tomato leaves, as shown in Fig. 4a and Supplementary Fig. 4. Figure 4a apparently showed the uniform deposition of TEB NPs (-120–140 nm) on the leaf. There was no obvious aggregation and the particles were uniformly distributed throughout the leaf with a narrow particle size distribution. In contrast, for SC and WP on the tomato leaves, there were obviously large aggregates or chunks of more than 400 nm to microns.

The efficient wettability and retention of pesticides on the surface can greatly improve the utilization of pesticides. A small contact angle can prevent droplets from rolling off the leaf surface. The hydrophilic tomato leaves and the hydrophobic wheat leaves were chosen to evaluate the wettability of the formulations. As shown in Fig. 4b, the TEB NPs had a much smaller contact angle than commercial SC and WP on both the tomato and wheat leaves. The assembly of amphiphilic block copolymer PDMAEMA-*b*-PCL made it more effective in covering and protecting the TEB hydrophobic core. As a result, the size of TEB NPs prepared by FNP was smaller and the size distribution was narrower than those of commercial formulations. Therefore, the obtained NPs exhibited a better dispersion in aqueous solutions and

permeability, which were important for their uniform distribution and diffusion on the leaf surface.

As shown in Fig. 4c, the liquid holding capacity (LHC) was measured to evaluate the wettability of the formulations, which directly affected the accumulation of active ingredients on leaves. The higher LHC of commercial SC and WP than that of water was likely due to a large number of surfactants and organic solvents in their formulations. The LHC of the block copolymer (BCP) solution was higher than that of water, which may be due to the hydrogen bonding between the amino group of BCP and the leaves. However, the TEB solution cannot well wet the leaves, so the variation of LHC between the TEB solution and water could be negligible. Compared to commercial formulations (SC and WP), the TEB NPs showed the highest LHC. This was due to the abundant functional groups on PDMAEMA-*b*-PCL forming hydrogen bonds with fatty acids, alcohols, and aldehydes on the waxy layer of the leaves, improving the wettability of TEB NPs on the leaves, and achieving the excellent wetting effect of commercial formulations without the use of surfactants[4].

Fluorescence microscopy was used to compare the adhesion and washing resistance of TEB NPs, SC, WP, and free TEB solution as shown in Fig. 4d, e and Supplementary Figs. 5–6. As shown in Supplementary

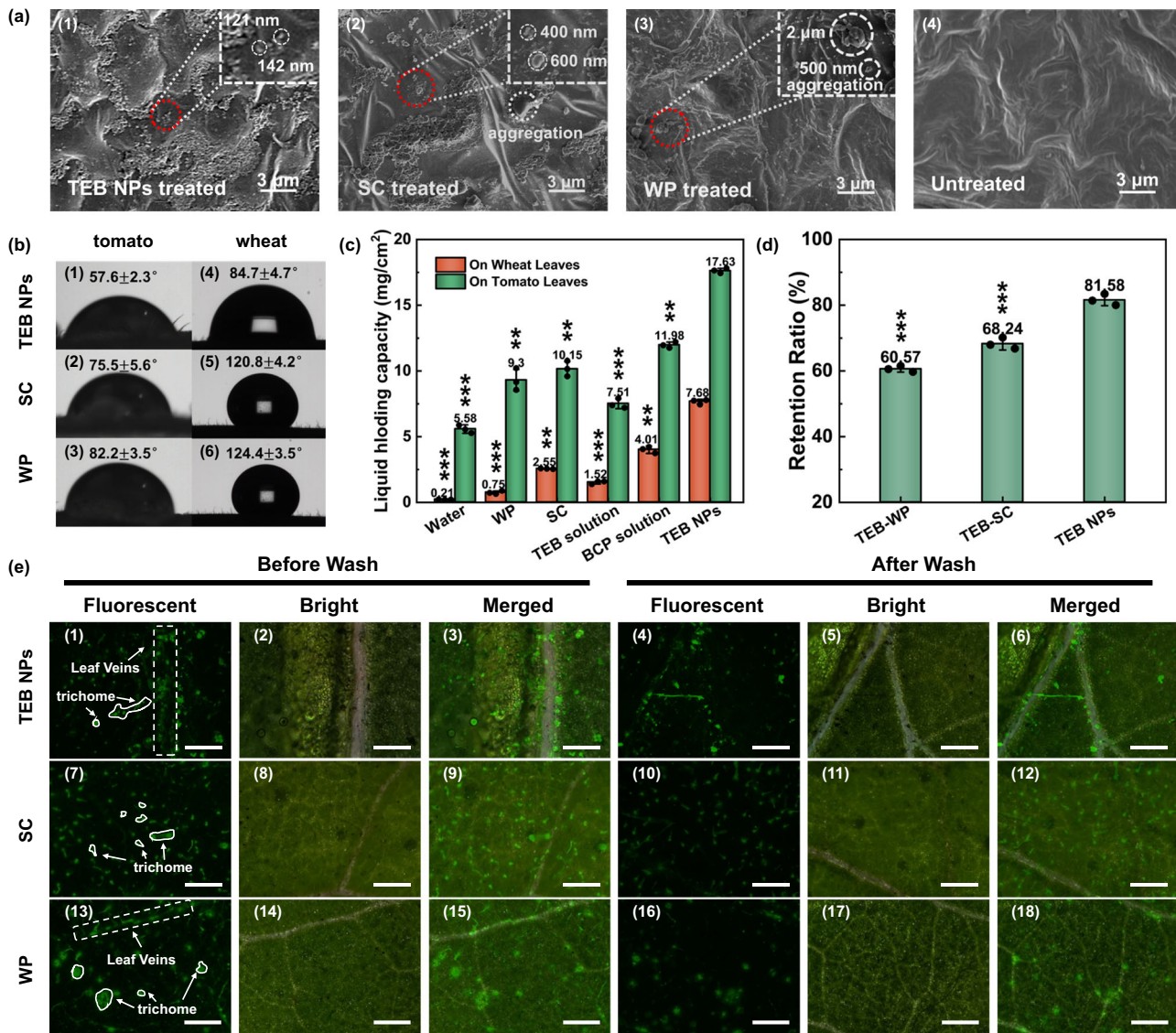

**Fig. 4 | Wetting and foliar affinity of nanopesticides with different formulations. a** SEM images of tomato leaves coated with different formulations (1) TEB NPs, (2) SC, (3) WP, and (4) untreated (each experiment was repeated three times independently with similar results). **b** Contact angle images of (1) TEB NPs, (2) SC, and (3) WP on the tomato leaves, (4) TEB NPs, (5) SC, and (6) WP on the wheat leaves. **c** The liquid holding capacities (LHC) of tomato leaves and wheat leaves after dipping in different solutions ($n = 3$ independent experiments, Data are presented as mean values ± SD). **d** Retention ratio for the series of fluorescence images of different TEB formulations ($n = 3$ independent experiments, Data are presented as mean values ± SD). **e** Fluorescence images of TEB NPs, SC, and WP on the tomato leaves to mimic rain washing resistance (Scale bar is 200 μm, each experiment was repeated three times independently with similar results). TEB NPs (1–3) before and (4–6) after wash; SC (7–9) before and (10–12) after wash; WP (13–15) before and (16–18) after wash. Statistical significance was defined by one-sample $t$-test (two-sided, **$P < 0.01$, ***$P < 0.001$, vs TEB NPs group).

Fig. 5, the fluorescence intensity of dialysate was much lower than that of commercial SC and WP diluted at the same ratio as the dialysate, which indicated that FITC was co-formulated by SC and WP. After washing with water, the fluorescence of the TEB solution shown in Supplementary Fig. 6 almost disappeared, and approximately 32.47% of the coverage area was retained. However, TEB NPs maintained a strong fluorescence (Fig. 4e, 1–6), which had 88.58% of the coverage area retained especially on leaf veins, trichrome, and epidermal, showing that the strong adhesion of the carrier PDMAEMA-*b*-PCL can effectively prevent the pesticide from being washed away by rain. The wax layer on the leaf surface is composed of higher fatty acids, alcohols, and aldehydes, and the functional groups on PDMAEMA-*b*-PCL can hence form robust hydrogen bonds with leaves. In addition, positively charged PDMAEMA-*b*-PCL interacted with the negatively charged group on the leaf surface via electrostatic attraction to improve the deposition efficiency of the pesticide on the target

surface[4]. Keeping the formulations on the waxy epidermis of plants and resisting rain erosion can increase the exposure and uptake of active ingredients by plants. Therefore, it is possible to use a low dosage of pesticide to achieve a good fungicidal effect.

Global agricultural productivity is increasingly compromised by climatic constraints, such as heat and flooding. Therefore, NPs that have excellent adhesion at high temperatures and maintains good resistance to heavy rain washouts can make modern agriculture more adaptable to extreme weather changes. The tomato leaves were coated by TEB NPs at different temperatures (30, 40, and 50 °C) to evaluate the wettability of NPs via FNP at high temperatures and rain. As shown in Fig. 5a and Supplementary Fig. 3, the images of SEM showed that TEB NPs had smaller sizes at high temperatures. As seen in Fig. 5b and Supplementary Fig. 7, the contact angle at higher temperature (50 °C) was 108.6°, which was significantly higher than that at low temperature (30 °C, 80.7°). This

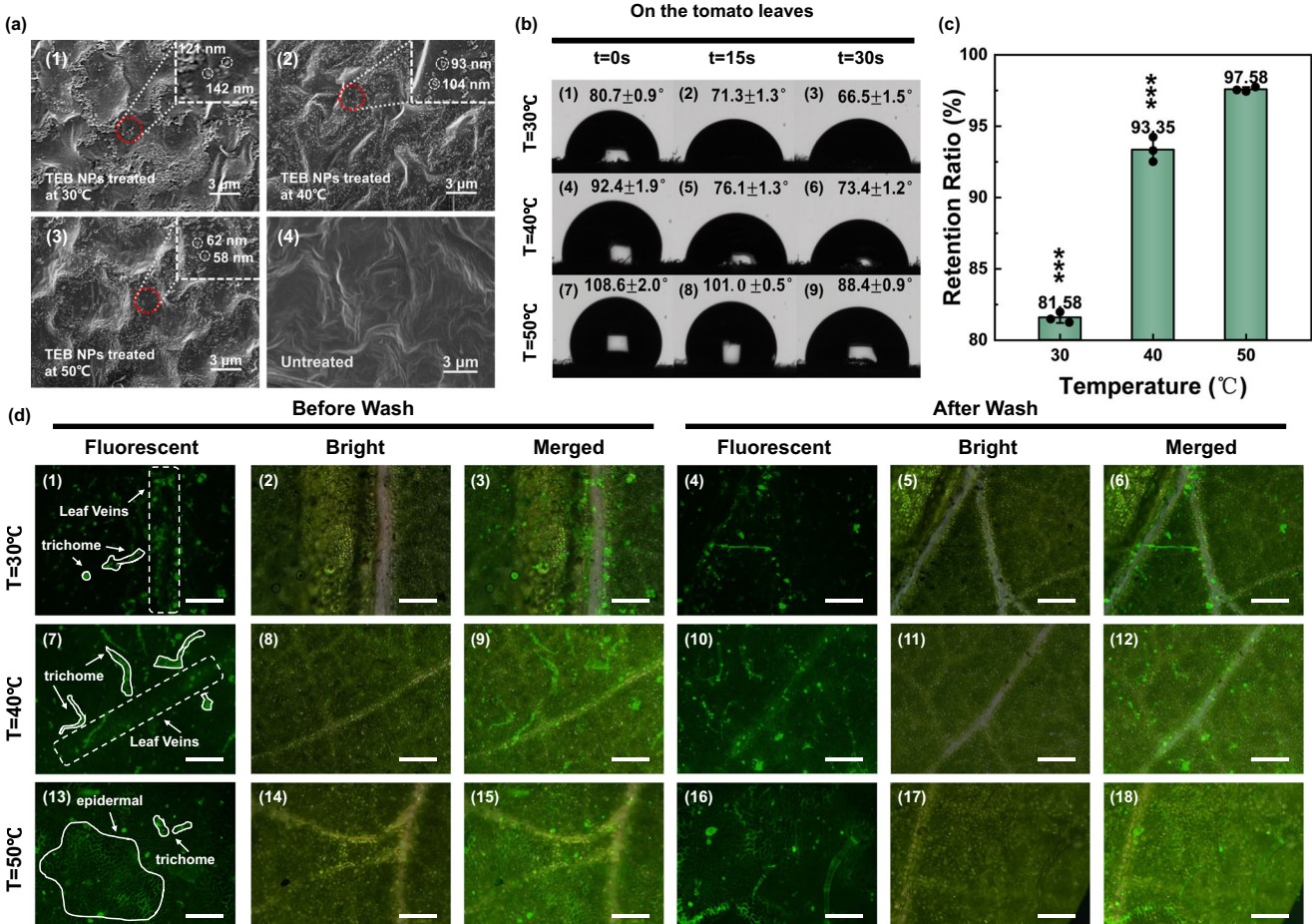

**Fig. 5 | Wetting and foliar affinity of TEB NPs with different temperatures. a** SEM images of tomato leaves coated TEB NPs at (1) 30 °C, (2) 40 °C, (3) 50 °C, and (4) untreated (each experiment was repeated three times independently with similar results). **b** Contact angle images of the tomato leaves coated by TEB NPs at (1–3) 30 °C, (4–6) 40 °C, and (7–9) 50 °C within 0, 15, and 30 s. **c** Retention ratio for the series of fluorescence images of TEB NPs at different temperatures (*n* = 3 independent experiments, Data are presented as mean values ± SD). **d** Fluorescence images of TEB NPs at the different temperatures on the tomato leaves to simulate rain washing resistance (Scale bar is 200 μm, each experiment was repeated three times independently with similar results). TEB NPs (1–3) before and (4–6) after wash at 30 °C, (7–9) before and (10–12) after wash at 40 °C, (13–15) before and (16–18) after wash at 50 °C. Statistical significance was defined by one-sample *t*-test (two-sided, **P < 0.01, ***P < 0.001, vs 50 °C group).

meant that the TEB NPs-loaded tomato leaves, dried at higher temperatures, had a higher contact angle with rainwater and allowed the rainwater to roll off the leaf surface more efficiently, making the NPs on the leaves less likely to be washed away. Figure 5c, d) showed that after washing with water, the fluorescence of NPs at 30 °C (Fig. 5d, 1–6) disappeared to some extent, and approximately 81.6% of the coverage area was retained. However, NPs at 40 °C and 50 °C (Fig. 5d, 13–18) maintained a stronger fluorescence, with 93.4% and 97.6% of the coverage area retained, respectively, demonstrating that the much better adhesion of the NPs at higher temperatures can effectively prevent the pesticide from being washed away by rain. The thermosensitive polymer PDMAEMA-*b*-PCL has a low critical solubility temperature (LCST) of ~40 °C and is hydrophilic below the LCST temperature but hydrophobic above LCST. At high temperatures, the copolymer shell collapses, resulting in a reduction of the size of TEB NPs. The small size effect of nanomaterials can reduce foliar pesticide shedding and improve pesticide utilization. The smaller the particle size was, on the other hand, the greater the contact area with the leaf was, which reinforces the interaction between the NPs and the leaves, thereby improving the adhesion. TEB NPs performed better when used in warm and humid environments than in dry environments. Therefore, in the heat and rainy environments, the leaves coated

with NPs and dried at high temperatures had greater adhesion and washing resistance, which was beneficial to enhance the pesticide utilization efficiency.

### Biological assay of TEB NPs

To confirm the uptake and translocation of TEB NPs in fungus and tomato plants, the FITC-labeled NPs were used to visualize the translocation under CLSM. For better comparison, both blank and treated samples were observed under CLSM. As shown in Fig. 6a and Supplementary Fig. 8, TEB NPs were detected in fungi under CLSM, and no fluorescence appeared in blank samples. FITC-labeled NPs were used to track the distribution of the carriers in plants as shown in Fig. 6b. Similarly, there was no fluorescent signal in the blank tomato plants (Fig. 6b 7–9), while in treated plants, FITC-labeled NPs could be seen clearly in the tomato leaves under CLSM as shown in Fig. 6b 10–15. The fluorescent signal could be clearly seen on the leaf surface of the treated leaves (Fig. 6b 10–12), while the fluorescence signal in the leaf cells could also be seen from the untreated section of the leaf adjacent to the treated position, as shown in Fig. 6b 13–15. As shown in Supplementary Fig. 9, the fluorescence intensity of the tomato flesh portion was much lower than that of the leaf, indicating a low risk for human consumption. This finding is in agreement with the reports that NPs via the FNP method could enter the plant from the surface of

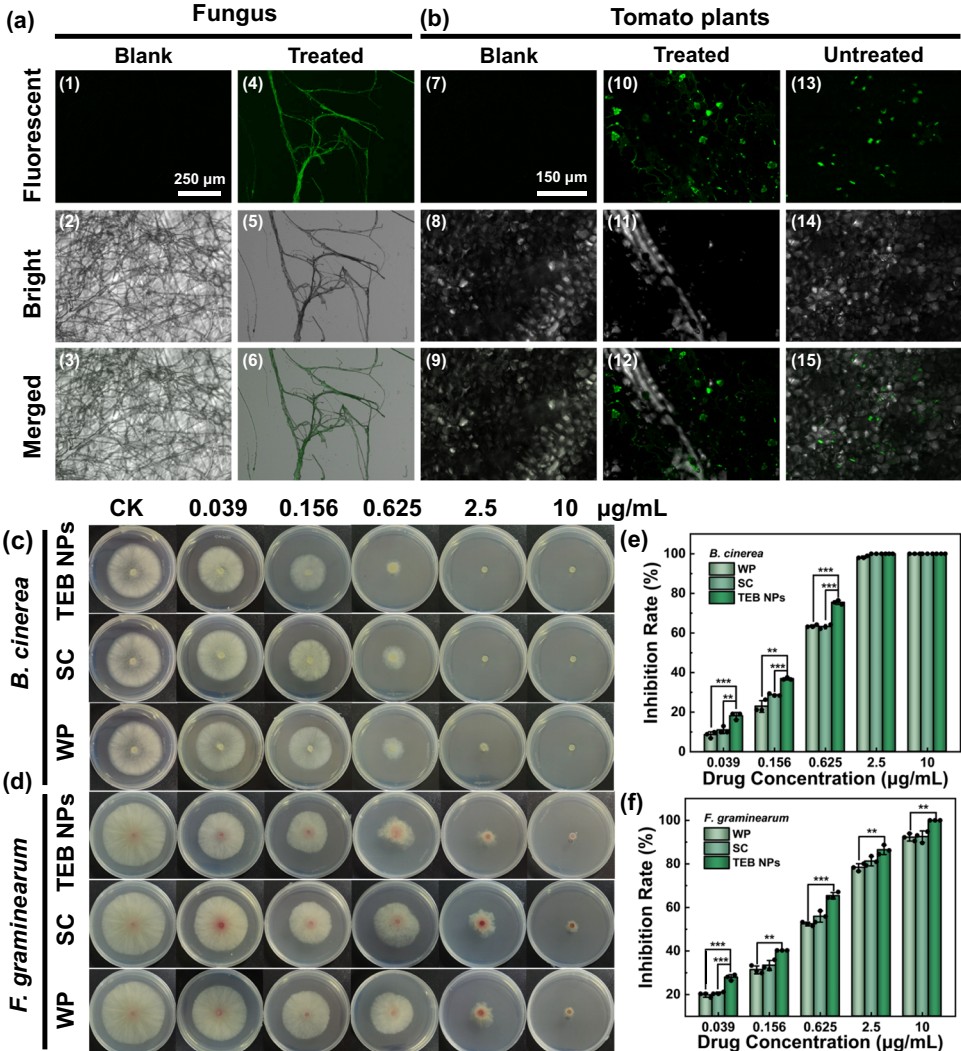

**Fig. 6 | Biological assay of TEB NPs.** CLSM images of **a** mycelia of tomato gray mold (*B. cinerea*), **b** leaves of tomato plants treated with TEB NPs (each experiment was repeated three times independently with similar results). (1–3) and (7–9) were blank samples, (4–6) were treated samples, (10–12) were images of leaves treated with FITC/TEB NPs, and (13–15) were images of the untreated section of the leaf adjacent to the treated position. Images of the fungicidal activity of TEB NPs, SC, and WP against **c** *B. cinerea* (tomato gray mold), and **d** *F. graminearum* (wheat fusarium head blight) after the 3d-treatment. Inhibition rate at different drug concentrations against **e** *B. cinerea* and **f** *F. graminearum* (*n* = 3 independent experiments, Data are presented as mean values ± SD). The concentration of TEB varied from 0.039, 0.156, 0.625, 2.5 to 10 µg/mL. Statistical significance was defined by one-sample *t*-test (two-sided, **$P < 0.01$, ***$P < 0.001$, vs TEB NPs group).

leaves and carry exogenous substances to other parts of the plant[20]. It indicated that the TEB NPs via the FNP could serve as a carrier to deliver pesticides in tomato plants and fungus.

TEB is one of the triazole therapeutic fungicides developed by Bayer Corporation that typically has three functions: protection, treatment, and eradication. It is highly effective, broad-spectrum, systemic, and has low toxicity, and is widely used for controlling plant diseases such as powdery mildew and fusarium head blight[35]. The block copolymer PDMAEMA-*b*-PCL used in this study has excellent biocompatibility (Supplementary Fig. 10), biodegradability, and numerous bioactivities, and is non-toxic[27,36–38].

The antifungal properties of TEB NPs were tested against *B. cinerea* and *F. graminearum* as indicator organisms. The concentrations of pesticide formulations required to show 50% of their maximal effect ($EC_{50}$) were determined. To better demonstrate the enhanced antifungal efficiency of TEB NPs, SC and WP were also used for comparison. The results are shown in Fig. 6c–f and Supplementary Fig. 11, and the $EC_{50}$ values of TEB NPs, SC, and WP were summarized in Table 1. The $EC_{50}$ of SC (0.672 and 0.712 µg/mL for *B. cinerea* and *F. graminearum*) was slightly lower than that of WP after 3 days of

inoculation, and the resulting $EC_{50}$ values of WP were 0.757 and 0.873 µg/mL, respectively, while the difference was insignificant. It is worth noting that the $EC_{50}$ of TEB NPs (0.339 and 0.344 µg/mL for *B. cinerea* and *F. graminearum*) was significantly lower than that of commercial TEB (SC and WP). Due to the limited volume of a single cell and its ability to internalize a certain volume of NPs, higher EE and smaller size were expected to bring higher biological efficacy[22]. As shown in Table 1, the NPs via FNP had the smallest size than commercial TEB, and the EE of NPs was up to approximately 90%, resulting that TEB NPs had much greater biological activity and lower $EC_{50}$ than the other two formulations.

Tomato leaves and wheat coleoptile were used as preserved organisms and treated, respectively, by (a) TEB NPs, (b) suspension concentrate (SC), (c) wettable powder (WP), and (d) water. The protective and curative effects of these treatments were observed 5 days later, as shown in Fig. 7. For the first 4 days, Supplementary Figs. 12–15 showed the protective activity on tomato leaves, and the curative activity was displayed in Supplementary Figs. 16–19. Similarly, for wheat coleoptile, Supplementary Figs. 20–23 showed protective activity, and Supplementary Figs. 24–27 showed curative

**Table 1 | The inhibition of *B. cinerea* and *F. graminearum* by using various TEB suspensions**

| Sample | Particle size (nm) | PDI | EC$_{50}$ (99% confidence interval) μg/mL | |
|---|---|---|---|---|
| | | | *B. cinerea* | *F. graminearum* |
| TEB NPs | 118 ± 4 | 0.24 ± 0.02 | 0.339 (0.244–0.470) | 0.344 (0.224–0.531) |
| SC | 882 ± 10 | 0.366 ± 0.07 | 0.672 (0.467–0.965) | 0.712 (0.473–1.071) |
| WP | 4169 ± 54 | 0.642 ± 0.1 | 0.757 (0.545–1.051) | 0.873 (0.580–1.315) |

activity. Regarding the protective activity, the control efficacy of TEB increased with the increase in the dosage used on tomato leaves, and TEB NPs outperformed the commercial TEB formulations with 97% control efficacy at the concentration of 100 μg/mL, while commercial TEB, respectively, had 83% and 79% for SC and WP at the same concentration as illustrated in Fig. 7a, b. In addition, for curative activity, as shown in Fig. 7c, d, the control efficacy for 100 μg/mL commercial TEB reached 86% and 71%, respectively, lower than that for 100 μg/mL TEB NPs (96%). TEB NPs had the same significant effect on wheat coleoptile as on tomato leaves. Figure 7e, f showed the protective activity, and TEB NPs at 20 μg/mL had 83% control efficacy, which was obviously higher than SC (70%) and WP (55%). Furthermore, for curative activity on wheat coleoptile, as shown in Fig. 7g, h, TEB NPs had a better performance (88%) than SC (78%) and WP (64%) at a concentration of 100 μg/mL. After the TEB formulation was sprayed on the plant leaves, the NPs coated on the surface prevented organisms from being infected by fungus. The small particle size, great adhesion performance, and high encapsulation efficiency strongly guarantee that TEB NPs via the FNP method demonstrate excellent performance on protective and curative activity for both tomato and wheat coleoptile.

## Safety assessment of TEB NPs

When zebrafish larvae were exposed to TEB formulations, their survival rate decreased with increasing the concentrations of the pesticides, as shown in Fig. 8a and Supplementary Figs. 28–33. The LC$_{50}$ value of TEB NPs to zebrafish larvae was 12.813 μg/mL, while commercial TEB, respectively, had 4.583 μg/mL and 0.449 μg/mL for WP and SC at the same concentration, suggesting that TEB NPs had a high biosafety level, as shown in Table 2 and Supplementary Table 2. According to the LC$_{50}$, the acute toxicity of TEB NPs, WP, and SC to zebrafish can be defined as low (>10 μg/mL), moderate (1.0–10 μg/mL), and high toxicity (0.1–1.0 μg/mL), respectively[39,40]. The acute toxicity of TEB NPs to zebrafish showed a more than 25-fold reduction as compared to that of SC, which may ascribe to the encapsulation of the active ingredient, and the absence of organic solvents in TEB NPs. The results showed that the safety of TEB NPs to aquatic organisms was much higher than that of commercial TEB.

During the production and application process, pesticides can easily enter the operators' bodies, for example by inhalation, therefore, cell viability as a proxy for in vitro toxicity is an indicator of the safety of pesticides. As shown in Fig. 8b, the survival rate of BEAS-2B cells (Bronchial Epithelium transformed with Ad12-SV40 2B) treated with 1 μg/mL TEB NPs was 97% after 24 h of incubation, whereas that of BEAS-2B cells treated with SC was only 77% at the same concentration. the cell viability for 50 μg/mL SC reached 29%, respectively, lower than that for 50 μg/mL TEB NPs (43%). This result showed that TEB NPs exhibited good biocompatibility and had a smaller effect on BEAS-2B cell viability. Explanations for this difference could be that encapsulating TEB into nanopesticides reduces the chance of being exposed to high drug concentrations and that the formula of commercial SC may contain unknown additives[40,41].

In this work, a nanopesticides formulation (TEB/PDMAEMA-*b*-PCL NPs) with effective deposition and water repelling was prepared by flash nanoprecipitation (FNP), a rapid and efficient

nanoparticle precipitation method, with the charged and temperature-responsive block copolymer PDMAEMA-*b*-PCL. The size of NPs was controlled by varying the copolymer concentration and flow rate during the FNP process. Compared with NPs via the thermal dynamic assembly method (TM), NPs prepared by FNP displayed smaller sizes, less polydispersity, improved stability and most importantly better controlled surface properties. TEB NPs have good stimuli-release properties, thermal stability, and photostability. NPs via FNP showed better leaf adhesion, effective deposition, and water repelling than commercial TEB (SC and WP). Meanwhile, higher contact angle and the coverage area retained demonstrated that the much better adhesion of the NPs at higher temperatures (50 °C) which can effectively prevent the pesticide from being washed away by rain. In addition, the inhibition rate and the antifungal activity of TEB NPs on *B. cinerea* and *F. graminearum* without the use of surfactants and organic solvents were higher than commercial TEB (SC and WP) by the PDA culture medium assay, the protective and curative assays on tomato leaves and wheat coleoptile. Compared with commercial SC and WP, TEB NPs had a smaller effect on Zebrafish survival rate and BEAS-2B cell viability, which exhibited low potential environmental toxicity and risks for human consumption. Such NPs system is expected to provide a simple and rapid method for the application of TEB, which can enhance pesticide droplet deposition and adhesion, improve pesticide adaptability to extreme weather conditions, and maximize the use of pesticides, tackling one of the application challenges of modern pesticides.

## Methods
### Ethics statement
All zebrafish experiments were conformed to the Zebrafish Information Network guidelines for the care and use of laboratory animals and ethically approved by the Laboratory Animal Ethical Committee of East China University of Science and Technology (Protocol number 2006272).

### Materials
Tebuconazole (TEB, 97.5%) was provided by Qizhou Green Chemical Co. Ltd. (Jiangsu, China). TEB suspension concentrate (SC, technical grade) with a 43% mass ratio was obtained from Bayer (China) Co. Ltd. TEB wettable powder (WP, technical grade) with an 80% mass ratio was obtained from Shanghai Shengnong Pesticide Co. Ltd. Poly-(2-(dimethylamino)ethylmethylacrylate)-*b*-poly(ε-caprolactone) (7k-*b*-8k, PDMAEMA-*b*-PCL) was synthesized in a previous work[42]. Fluorescein isothiocyanate isomer (FITC) was purchased from Shanghai Macklin Biochemical Co. Ltd. Tetrahydrofuran (THF) and ethanol anhydrous were obtained from Shanghai Adamas beta Co. Ltd. The dialysis membranes were purchased from Shanghai Yuanye Biotechnology Co. Ltd. Cell-culture products were purchased from Keygentec (Nanjing, China). All the reagents were analytical grade unless otherwise stated, and ultra-pure water was used in all experiments. Ultra-pure water was obtained by a Milli-Q water purification system.

The susceptible *Botrytis cinerea* (a pathogen that causes gray mold) was collected from tomato fields, and *Fusarium graminearum* (a pathogen that causes fusarium head blight) was collected from

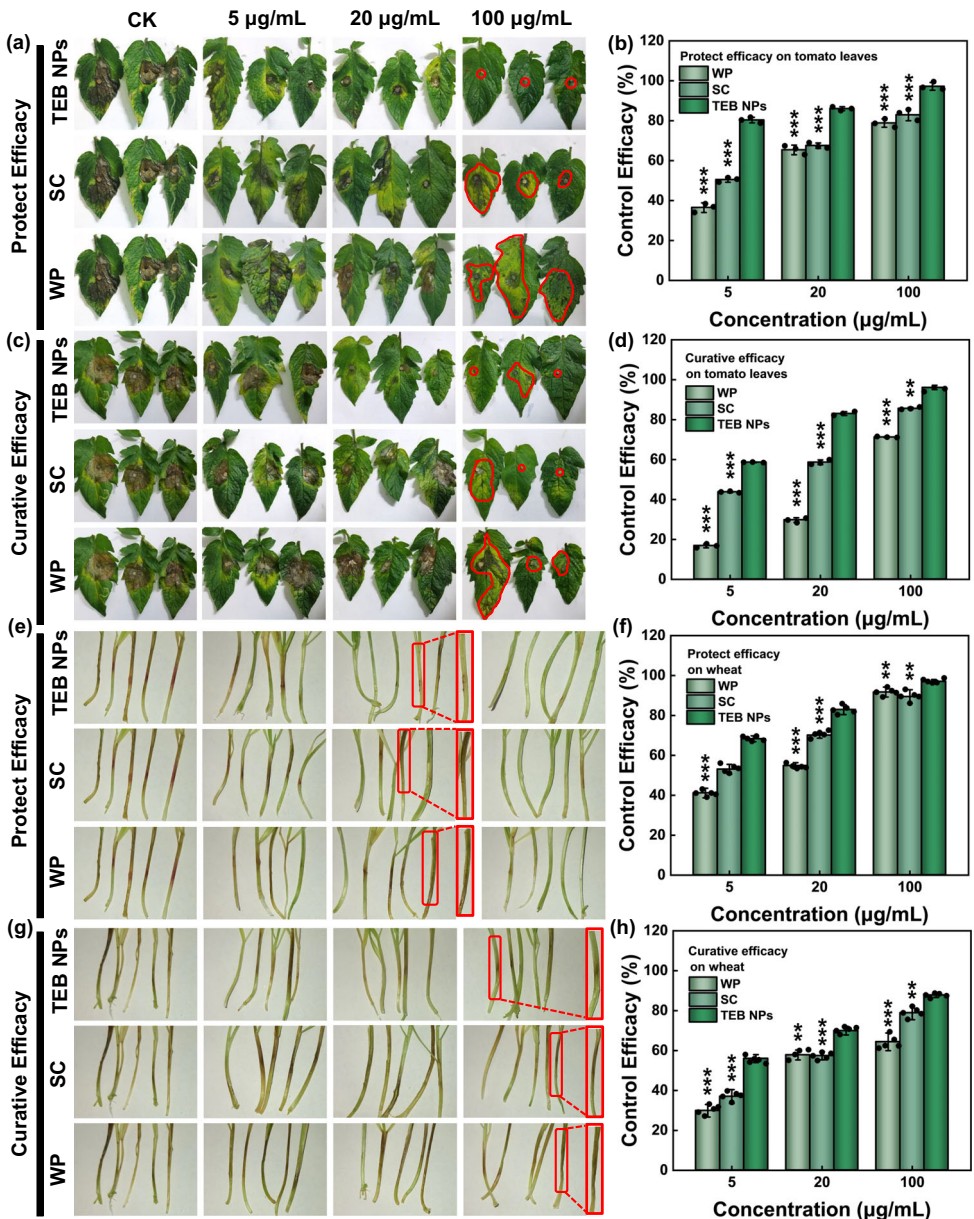

**Fig. 7 | Protective and curative efficacy of different TEB formulations.**
**a** Protective activities and **b** control efficacy ($n = 3$ independent experiments. Data are presented as mean values ± SD) of TEB NPs, SC, and WP on tomato leaves. The concentration of TEB varied from 5, 20 to 100 μg/mL. **c** Curative activities and **d** control efficacy ($n = 3$ independent experiments. Data are presented as mean values ± SD) of TEB NPs, SC, and WP on tomato leaves. **e** Protective activities and **f** control efficacy ($n = 3$ independent experiments, Data are presented as mean values ± SD) of TEB NPs, SC, and WP on wheat coleoptile. **g** Curative activities and **h** control efficacy ($n = 5$ independent experiments. Data are presented as mean values ± SD) of TEB NPs, SC, and WP on wheat coleoptile. Statistical significance was defined by one-sample $t$-test (two-sided, **$P < 0.01$, ***$P < 0.001$, vs TEB NPs group).

wheat fields. They were identified, isolated from agricultural locations within Shanghai suburbs in the past, and conserved by our laboratory. The fungi was grown on potato dextrose agar (PDA) at $25 \pm 2\,°C$ in routine assays to assess the antifungal action of TEB in vitro and in vivo.

### Preparation of TEB NPs

TEB/PDMAEMA-*b*-PCL NPs (TEB NPs) were prepared using the FNP method based on a confined impinging jet with dilution (CIJ-D), which was shown in Fig. 1. In the formulation process, 2.0 mL of THF with dissolved 2.0 mg of TEB and the amphiphilic copolymer PDMAEMA-*b*-PCL was loaded in one gas-tight plastic syringe (20 mL, Kangli Medical Products, China). PDMAEMA-*b*-PCL was dissolved in THF with TEB:PDMAEMA-*b*-PCL mass ratio of 0.5:1, 1:1, 2:1, 4:1, and 8:1. Another syringe was loaded with 2.0 mL of water. The two

syringes with equal volume were pushed rapidly by a pump (Harvard Apparatus, PHD 2000 programmable) to inject the liquids into the chamber of the CIJ-D mixer at equal flow rates, where the two streams were instantaneously mixed. Then the fluid came out from the outlet tubing and fell into a beaker which was pre-filled with 16.0 mL of water for a dilution. The samples were dialyzed against ultra-pure water for 24 h to remove organic solvents and free TEB using a dialysis bag with a 3.5k Da molecular-weight-cut-off (MWCO) membrane and then stored at room temperature. All the treatments were performed three times.

TEB NPs were prepared by conventional thermal dynamic assembly method (TM) as follows: TEB (1 mg/mL) in THF was added into a solution of PDMAEMA-*b*-PCL (1 mg/mL) in THF. Then ultra-pure water with a water/THF volume ratio of 9:1 was added to the as-prepared mixture solution under vigorous stirring.

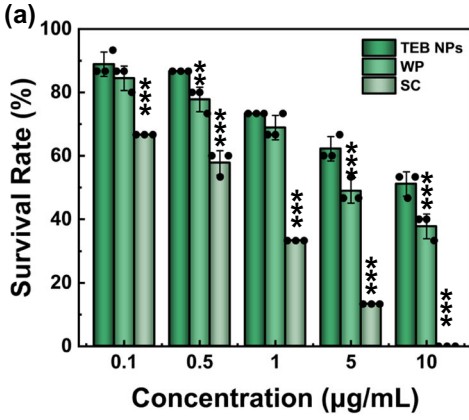

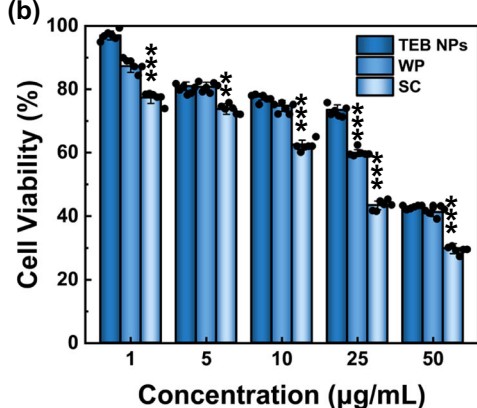

**Fig. 8 | Safety assessment of TEB NPs. a** The survival rate of zebrafish larvae (5 dpf) exposed to different TEB formulations with concentrations of 0.1, 0.5, 1, 5, and 10 µg/mL at 96 h ($n = 15$ larvae examined over 3 independent experiments, Data are presented as mean values ± SD). **b** Cell viability of different TEB formulations with concentrations of 1, 5, 10, 25, and 50 µg/mL after 24 h of incubation ($n = 1 \times 10^4$ cells examined over 6 independent experiments, Data are presented as mean values ± SD). Statistical significance was defined by one-sample $t$-test (two-sided, **$P < 0.01$, ***$P < 0.001$, vs TEB NPs group).

## Characterization of TEB NPs

The average hydrodynamic diameter and polydispersity index (PDI) of TEB NPs were measured with a NICOMP 380 ZLS at 0 °C, 25 °C, 38 °C, and 54 °C with a scattering angle of 90° multiple times[43]. The average of three measurements was reported. The size of SC, and WP were measured with a NICOMP 380 ZLS with a scattering angle of 90° multiple times.

The dried samples were ground into a fine powder with potassium bromide (KBr) and pressed into thin slices and scanned 32 times within the wavenumber range of 400 to 4000 cm$^{-1}$ through a Spectrum 100 Fourier transform infrared spectrometer (Perkin Elmer Inc., USA).

The morphology of NPs was observed on a JEOL JEM-1400 TEM instrument with an acceleration voltage of 200 kV. One drop of the TEB NPs solution was deposited on a carbon-coated copper grid. The droplet was allowed to dry at room temperature.

Zeta potentials (from electrophoretic mobility) of NPs were measured using a Malvern Zetasizer Nano ZEN3700 (Malvern Instruments, Worcestershire, UK). The pH of the sample solution was fixed at 6.0 and three replicate measurements per sample were performed.

TGA simultaneous thermal analyzer (STA 449 F3) was used for thermogravimetric analysis to record the weight-loss data of the material with temperature changes. The temperature was increased from 30 °C to 600 °C at a heating rate of 10 °C/min and a nitrogen flow rate of 20 mL/min.

## Encapsulation efficiency and drug loading capacity

The encapsulation efficiency (EE) and drug loading capacity (DLC) of TEB NPs were determined by a UV-1900i UV-Vis spectrophotometer. After the removal of organic solvents and free TEB by dialysis, UV-Vis absorption spectra of samples that were diluted in THF were measured, and the concentration was determined from the absorbance at 269 nm using a standard curve A = 1.6534C where A is the absorbance and C is the concentration ($R^2 = 0.9999$, Supplementary Fig. 34). It is worth noting that PDMAEMA-$b$-PCL and solvents had almost no effect on TEB quantification using a UV-Vis spectrophotometer.

### Table 2 | The LC$_{50}$ values of different TEB formulations to Zebrafish larvae

| Sample | LC$_{50}$ (µg/mL) | 99% confidence interval |
|---|---|---|
| TEB NPs | 12.813 | 6.342–25.888 |
| WP | 4.583 | 2.848–7.375 |
| SC | 0.449 | 0.337–0.598 |

The EE was determined as the difference between the amount of TEB in the TEB NPs and the total amount of added TEB:

$$EE(\%) = \frac{\text{amount of encapsulated TEB}}{\text{total amount of added TEB}} \times 100\% \qquad (1)$$

The DLC was defined as the ratio of encapsulated TEB in NPs to the total mass of the NPs:

$$DLC(\%) = \frac{\text{amount of encapsulated TEB}}{\text{total amount of NPs}} \times 100\% \qquad (2)$$

## Stimuli-response release behavior of TEB NPs

To assess the release profile of TEB from PDMAEMA-$b$-PCL, release experiments were performed at different temperatures (30, 40, and 50 °C) and pH values (4.5, 6.0, and 7.4). TEB NPs (3 mL) were placed into a dialysis bag (MWCO, 3500 Da), which was placed in 27 mL of dialysate containing ethanol and water (20:80, v/v) with agitation at 200 rpm. At specific time points, 3 mL of the dialysate was taken out, and then the same volume of fresh dialysate was added to ensure the same volume of the medium. The TEB concentration was determined using a UV-Vis spectrophotometer (269 nm) using a standard curve A = 1.250 C ($R^2 = 0.9998$, Supplementary Fig. 35). All the treatments were performed three times.

## Photostability performance of TEB NPs

The TEB NPs and free TEB were, respectively, prepared with a TEB concentration of 0.3 mg/mL in THF/water solution (1090, v/v). 5 mL of TEB NPs, free TEB, SC, and WP with the same concentration were placed into the quartz pools with 60 mm diameter. The quartz pool was placed in a dark cabinet equipped with a UV lamp at a distance of -10 cm from the light source ($E_{max} = 254$ nm, 16.4 W/m²). After irradiation of 0.5, 1, 2, 4, 6, 9, and 12 h, 0.3 mL of the solution was retrieved and diluted with THF for analysis of the UV-Vis spectra.

The degradation process was studied by the pseudo-first-order kinetic model, according to the following equation:

$$\ln \frac{C_n}{C_0} = -kt \qquad (3)$$

$$t_{\frac{1}{2}} = \frac{\ln 2}{k} \qquad (4)$$

where $C_n$ (mg/mL) is the degradation concentration of TEB at different times, $C_0$ is the initial concentration of TEB, and $k$ ($h^{-1}$) is the degradation kinetic constant.

## Wettability and adhesion properties on foliage

The contact angle was used to evaluate the wettability of the formulations. Fresh tomato and wheat leaves cultivated in the laboratory were selected and carefully washed several times with ultra-pure water to remove dust from the surface ensuring that the surfaces of the leaves were not damaged. After dying at room temperature, the tomato and wheat leaves were fixed on the surface of a glass slide, and 15 μL of different solutions were dropped on the surface of the leaves using a micro-syringe. The contact angle was captured by a contact angle meter (JY-82A, Chengde Digital Technology Apparatus) after 30 s. Each solution was applied three times on different leaves.

The tomato leaves were used to evaluate the adhesion of TEB NPs at different temperatures. Leaves were immersed in the solution for 20 s, dried at different temperatures (30, 40, and 50 °C), and then captured the contact angle of the coated leaves to assess the adhesion property. All the treatments were performed three times.

The dried tomato leaves were cut into $2 \times 2$ cm²-sized specimens, while the wheat leaves were cut into $1 \times 4$ cm², and immersed into 0.1 mg/mL TEB solution, PDMAEMA-*b*-PCL solution, TEB NPs, SC, and WP solutions. After 20 s, the leaves were lifted vertically with tweezers until there were no drops, and the mass of the leaves was weighed using an analytical balance (ME204E, Mettler Toledo). All tests were performed three times. The liquid holding capacity (LHC) was calculated using the following equation:

$$LHC = \frac{M_1 - M_0}{S} \qquad (5)$$

where $M_0$ and $M_1$ represent the weights of the leaves before and after soaking, and S is the surface area of the leaves.

In this study, the rain-fastness of different formulations on the tomato leaves surfaces was tested using a laboratory-scale wash-off method, and FITC was used for visibility[44].

The preparation of FITC-loaded NPs was performed according to the following experimental procedure. In the organic phase, 0.2 mg of FITC was added to prepare FITC-loaded TEB NPs. SC, and WP were dispersed individually in 20 mL of water, and then 0.2 mg of FITC was added into the suspensions and stirred at 500 rpm for 24 h. FITC-SC and FITC-WP (2 mL of each) were placed into a dialysis bag (MWCO, 3500 Da), which was placed in 18 mL of dialysate (ultra-pure water) with agitation at 200 rpm. After 1 h, 3 mL of the dialysate was taken out and the fluorescence intensity was determined to indicate that FITC was linked to the non-encapsulated TEB. All experiments about the loading procedure of FITC were conducted at room temperature in the dark. It is worth noting that the concentration of FITC was the same in all formulations.

The four formulations (0.5 mL) were dropped on the tomato leaves to dry at room temperature, and the leaves were sprayed with 10 mL of ultra-pure water at 60° bevel. FITC-loaded TEB NPs solution was dropped on the tomato leaves to dry at different temperatures (30, 40, and 50 °C), which was used to evaluate the variation of adhesion ability with temperature. Three replicates of the wash-off test were performed. The change before and after spraying was observed with a fluorescence-inverted microscope (Olympus IX73). Finally, the series of images were processed using ImageJ software to determine the coverage of the fluorescent deposit. The image before washing was regarded as the value for the initial 100% coverage, and the subsequent images were quantified as a percentage by reference to the initial image.

## Bioactivity studies of TEB NPs

The FITC-loaded TEB NPs were used to determine the uptake of NPs by the plants and fungi. Briefly, mycelial explants (5 mm in diameter) of *B. cinerea* grown on PDA plates which were treated with FITC-loaded TEB NPs solution (0.1 mg/mL in water) were used to explore the uptake of NPs in fungi. After incubation for 3 days, mycelial tips (5 mm) from the edge of the colony were carefully excised for imaging observation using Confocal Laser Scanning Microscopy (CLSM, Leica TCS SP8) with a laser excitation wavelength of 488 nm and a emission wavelength of 525 nm. To study the uptake and translocation of NPs in plants, tomato plants were selected with nearly equal sizes, and a middle leaf was treated with 1 mL of FITC-loaded TEB NPs solution (0.1 mg/mL in water). On the day after treatment, the untreated section of the leaf adjacent to the treated position was sampled for imaging observation with CLSM. To study the nanopesticides residues in the edible parts of the crops, tomato plants were selected, and a middle leaf was treated with 1 mL of FITC-loaded TEB NPs. At 24 h after treatment, the untreated section of the leaf and the tomato flesh portion adjacent to the treated position was sampled for imaging observation with CLSM. All the treatments were performed three times.

To evaluate the improvement of the antifungal activity of TEB by using the FNP method, *B. cinerea* and *F. graminearum* was used as the target organism by the PDA culture medium assay. Various TEB suspensions were added to a culture medium of PDA in a petri dish (90 mm in diameter). The concentration of TEB varied from 0.039, 0.156, 0.625, 2.5 to 10 μg/mL. Mycelial plugs (5 mm in diameter) ($S_0$) were cut from the actively growing edges of a 5-days-old fungal colony and placed upside down onto the center of each dish. As a control, water replaced TEB suspensions to treat the PDA culture medium. All the dishes were covered and incubated at $25 \pm 1$ °C for 3 days in a biochemical incubator (SPX-250, Binglin Electronic Technology Apparatus), and then the diameter (S) of the grown colony was measured. It is worth noting that PDMAEMA-*b*-PCL had almost no effect on the antifungal activity, as shown in Supplementary Fig. 10.

Three replicate dishes were used per test. The inhibition rate (IR) was calculated using the following equation:

$$IR = \frac{C - S}{C - S_0} \times 100\% \qquad (6)$$

where C (mm) is a diameter of a growing colony in a control, and S (mm) is the diameter of treatment with TEB suspension. The median inhibitory concentration ($EC_{50}$) of TEB suspension was estimated via linear regression of the $\log_{10}$-transformed fungicide concentration versus the probit-transformed IR value. The less $EC_{50}$ value is, the greater the antifungal activity of the suspension is.

Commercial wheat seeds of Jimai 22 were soaked in tap water and placed in a dark incubator at 25 °C for 24 h. A 90 mm Petri dish with wetted filter paper was placed at the bottom, and the germinated wheat was placed on the filter paper. Tap water was poured to half the height of the wheat seeds and the Petri dish was gently covered with gauze and continued to be incubated in the dark for 4 days. Afterward, the wheat was planted evenly in the soil and incubated at 25 °C and 80% RH for 10 days with 8 h of light per day, and wheat of uniform growth was selected for subsequent experiments.

Various formulations (TEB NPs, SC, and WP) were used to evaluate the protective and curative efficacy of wheat coleoptile. 20 mL of different solutions containing 0.1% Triton X-100 had a concentration of TEB varied from 5, 20 to 100 μg/mL. Deionized water containing 0.1% Triton X-100 without TEB was used as a blank control. All the treatments were performed five times.

Protective activity: the coleoptile was dipped in the prepared agent for 3 s, dried, and then placed in a Petri dish to continue incubation at 25 °C in the dark for 24 h. After puncturing a wound in the middle of the coleoptile with a needle, an explant of *F. graminearum* was inoculated on the wound. Then the Petri dish was covered with two layers of gauze, kept moist (90% RH), and put into a 25 °C incubator to continue the dark culture. After incubation for 5 days, the length of lesions on the wheat was measured, and the protective activity was calculated.

Curative activity: firstly, by using a needle to puncture a wound in the middle of the coleoptile, the wound was inoculated with an explant of *F. graminearum*. After incubation for 24 h, the coleoptile was dipped in the prepared agent for 3 s, dried, and then continued to incubate under moisture (90% RH) and dark conditions at 25 °C for the next 5 days. The length of lesions on the coleoptile was measured, and the curative efficacy (CE) was calculated using the following equation:

$$CE = \frac{C - L}{C} \times 100\% \qquad (7)$$

where C (mm) is the length of lesions on the wheat coleoptile in a control, and L (mm) is the length of lesions on the wheat coleoptile treated with TEB suspension.

Live tomato leaves of Suhong were obtained from a farm in Shanghai. The protective and curative efficacy of various formulations (TEB NPs, SC, and WP) on tomato leaves was estimated according to the protocol as on wheat. All the treatments were performed three times. After incubation for 5 days, the area of lesions on the tomato leaves was measured.

### Safety assessment of TEB NPs

To evaluate the acute toxicity of TEB NPs to aquatic organisms, zebrafish were selected as model animal[39]. WT line wild-type zebrafish larvae were obtained from the Laboratory of Aquatic Animal Diseases of East China University of Science and Technology (Shanghai, China), and maintained with E3 medium according to the standard protocol[45,46]. For safety assessment in zebrafish, 5 d postfertilization (dpf), zebrafish larvae (15 larvae per treatment) were randomly picked and immersed in a six-well plate with different TEB formulations at the concentration of 0.1, 0.5, 1, 5, 10 μg/mL. During the exposure, all the fish were unfed. Zebrafish deaths were recorded at the indicated time points. Each exposure was repeated thrice.

In vitro cytotoxicity was measured by using methyl thiazolyl tetrazolium (MTT) assay. BEAS-2B cells (Bronchial Epithelium transformed with Ad12-SV40 2B) were cultivated in 96-well plates at a density of $1 \times 10^4/100$ μL containing DMEM at 37 °C and 5 % $CO_2$ for 12 h. Subsequently, the medium was next replaced by fresh medium containing different mass concentrations of TEB formulations (0, 1, 5, 10, 25, and 50 μg/mL). After 24 h, MTT (10 μL, 5 mg/mL) was added to each well and the plate was incubated for an additional 4 h at 37 °C under 5 % $CO_2$. After the addition of 100 μL DMSO, the assay plate was allowed to stand at room temperature for 20 min. A Tecan Infinite M200 monochromator-based multifunction microplate reader was used to measure the optical density OD 570 value (Abs) of each well with background subtraction at 490 nm. All the treatments were performed six times. cell viability was calculated using the following equation:

$$\text{Cell viability} = \frac{\text{mean Abs value of treatment group}}{\text{mean Abs value of control}} \times 100\% \qquad (8)$$

### Statistical analysis

Origin software (Origin 2023 Corporation, U.S.) was used for the graph plot. The data were the mean values of the three experiments, expressed as the means ± the standard deviations (SD). The statistical analysis was performed using one-sample *t*-test, and **$P < 0.01$, ***$P < 0.001$ were used to show statistical significance. The data not showing asterisks, reveal not significant. The median lethal concentrations ($LC_{50}$) and the median inhibitory concentration ($EC_{50}$) values were calculated by probit regression model.

### Reporting summary

Further information on research design is available in the Nature Portfolio Reporting Summary linked to this article.

## Data availability

The source data generated in this study are provided in the Source data file. Source data are provided with this paper.

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

## Acknowledgements

J.T. and Y.X. acknowledges the support of the National Key Research and Development Program of the International Scientific and Technological Innovation Cooperation Project among governments (2021YFE0100400) and the National Natural Science Foundation of China (22378126). We thank Dr. Dahai Yang from the Laboratory of Aquatic Animal Diseases of East China University of Science and Technology for sharing WT line wild-type zebrafish larvae.

## Author contributions

J.T., Q.H., N.R., A.A., A.B.K., and Y.X. conceived the idea and designed the experiments; J.T., X.T., Y.C., and Y.X. developed the methodology and analyzed the data; Y.W., Z.Z., and J.Y. assisted to collect the data and interpretations; J.T. and Y.X. wrote the papers.

## Competing interests

The authors declare no competing interests.
