## [Peer Review File · Nature Communications]

Deposition and water repelling of temperature-responsive nanopesticides on leavesEditorial Note: Parts of this Peer Review File on p6 and p8 have been redacted as indicated to remove third-party material where no permission to publish could be obtained.

REVIEWER COMMENTS

Reviewer #1 (Remarks to the Author):

Manuscript review of NCOMMS-23-14365 "Effective Deposition and Water Repelling of Temperature-Responsive Nano-pesticides on Leaves."

General Comments- This manuscript provides a simple and rapid method for the application of tebuconazole-loaded NPs using the flash nanoprecipitation method with the block copolymer poly-(2-(dimethylamino) methylmethacrylate)-b-poly(ϵ -caprolactone) (PDMAEMA-b-PCL) as the carrier. The nanopesticides were characterized by SEM, TEM, FTIR, and DLS. The release kinetics were performed at different temperatures (30, 40, and 50 °C) and pH values (4.5, 6.0, and 7.4). Also, the nanopesticides were studied according to their photostability performance, wettability and adhesion properties on foliage, uptake in plants and fungi, and protective and curative activity. The topic is timely, adds some information to our understanding of the potential effect and impacts of nanoscale carriers of bioactive compounds, and is within the scope of Nature Communication. The paper is well-written and clearly presented. The experiments were carefully conducted, and the data support the authors' conclusions. However, as currently written, I have several specific comments that prevent me from recommending this paper for publication in its current form. With a significant amount of effort, the authors may be able to address these concerns, and if that is done, the paper may be acceptable for publication.

1. In the abstract, I suggest revising the objective of the manuscript and adding more quantitative results; be as specific and detailed as possible for the reader.
2. In the introduction, when discussing key studies from the literature, be as detailed, specific, and quantitative as possible.
3. I suggest using Scheme 1 as Graphical Abstract.
4. I suggest that the TEM of Fig.1-i be performed at the same magnification as in Figure 1-h. Moreover, it is necessary to show the recrystallization process discussed by the authors.
5. The authors describe that "the surrounding water further restricted the hydrophobic active ingredient (TEB) in the core of the NPs." Could the TEB also be associated with the polymer wall? What type of nanocarrier was fabricated (nanosphere or nanocapsules)? I also suggest discussing this related to the active ingredient release profile information - "The similar initial release profiles at different pH values may be attributed to TEB near the NPs surfaces."
6. Rain-fastness wash-off tests: Fluorescence microscopy was used to compare the adhesion and washing resistance of TEB NPs, SC, WP, and free TEB solution. However, what guarantees that FITC is linked to the non-encapsulated TEB? Would the rain-washing process have only eliminated FITC, not the pesticide from the leaves in Figure S5? Please further discuss it.
7. The authors reported that the block copolymer PDMAEMA-b-PCL is excellent biocompatibility, biodegradability, and non-toxic. However, TEB-loaded PDMAEMA-b-PCL will have a new fate and behavior in the environment. Although the risk assessment of this nanopesticide was not studied in the manuscript (which is acceptable given the number of results presented). I would suggest that something related to the "ecotoxicological aspects" of these nanocarriers will be briefly discussed in the manuscript.
8. It needs to be clarified how the particle size (nm) of SC and WP was measured in Table 1.
9. I also suggest clarifying how the analyses related to the length of lesions were performed in the plants.

Reviewer #2 (Remarks to the Author):

This manuscript reports the synthesis of BCP-stabilized nanocrystals of hydrophobic antifungal molecules prepared by flash nanoprecipitation (FNP). The resulting kinetically stabilized nanoparticles were used as leaf-adhesive particles that release encapsulated drug molecules to the plant leaves. The adhesive nature of the nanoparticles relies on the stimuli-responsive properties of PDMAEMA polymer blocks. The adhesive nanoparticles could enhance the efficiency of drug delivery to plant leaves, which could result in the reduction of the volume of applications of

pesticides. Although this is a well-performed and comprehensive study, I am not convinced that this work brings enough new discoveries and insights to the readership of Nature Communications. I believe that this manuscript suits better to more specialized journals. I also have two technical questions. (1) Because PDMAEMA-b-PCL was used as a stabilizer, nanoparticles produced by the FNP method are primarily stabilized by PCL blocks. Then, how can the authors relate the increase in the release of pesticides with a collapse or expansion of PDMAEMA blocks? (2) PDMAEMA block shows temperature- and pH-sensitive properties when the polymer is in solution. How can the authors make sure that the applied nanoparticles are in a wet or solution-like state?

Reviewer #3 (Remarks to the Author):

The manuscript titled "Effective Deposition and Water Repelling of Temperature-Responsive Nano-pesticides on Leaves" demonstrated a nano-formulated tebuconazole prepared by flash nanoprecipitation, which had good stimuli-release properties, thermal stability, photostability, as well as better leaf adhesion, effective deposition and water repelling than commercial formulations. Although the topic is of great interest, the manuscript in its actual form is suffering from some issues that will need major revisions.

1. The measurement of particle size after two months of storage at room temperature does not fully reflect the stability of nano-formulation. The stability of the NPs should be evaluated under different temperature according to the general principles. Specifically, three parallel samples were prepared and sealed. One sample was stored at 0 ± 2 °C for 7 d, the other two samples were stored separately at 25 ± 2 °C and 54 ± 2 °C for 14 d, and the particle size and PDI of the samples were measured. The author may refer to the methods in the following literature: "Synthesis and characterization of a novel stimuli-responsive zein nano delivery system for the controlled release of emamectin benzoate, Wang et al., 2022, Environmental Science Nano, 9, 4411-4422."

2. Although the author mentioned that nanocarriers are non-toxic, since this nano-formulation can increase the deposition of pesticides on crop leaves and change the transfer of pesticide within crops, the author should consider whether pesticide residues will increase in the edible parts of the crops.

3. The author conducted statistical analysis of the experimental data, but in the methods section, no information on the software used for statistical analysis was provided.

4. The description of some results is incorrect. For example, in the second paragraph of section 2.2.1 Stimuli response release of TEB NPs, "The cumulative release obtained at 50°C is 94%, higher than 70% at 30°C and 43% at 40°C after 48 h.", which is not consistent with Figure 2 (a).

Response to Reviewers

We gratefully thank the editor and all reviewers for helpful remarks and constructive suggestions, which have significantly improved the presentation of the manuscript and enabled us to raise the quality of the manuscript. We have carefully considered all comments from the reviewers and revised the manuscript accordingly. The manuscript has also been double-checked, and the typos and grammar errors we found have been corrected. Point-by-point responses to all the comments are provided below, with revisions indicated. We hope our revised manuscript can be accepted for publication.

Response to Reviewer 1:

Reviewer #1: This manuscript provides a simple and rapid method for the application of tebuconazole-loaded NPs using the flash nanoprecipitation method with the block copolymer poly-(2-(dimethylamino)methylmethacrylate)-*b*-poly(ϵ -caprolactone) (PDMAEMA-*b*-PCL) as the carrier. The nanopesticide were characterized by SEM, TEM, FTIR, and DLS. The release kinetics were performed at different temperatures (30, 40, and 50 °C) and pH values (4.5, 6.0, and 7.4). Also, the nanopesticides were studied according to their photostability performance, wettability and adhesion properties on foliage, uptake in plants and fungi, and protective and curative activity. The topic is timely, adds some information to our understanding of the potential effect and impacts of nanoscale carriers of bioactive compounds, and is within the scope of Nature Communication. The paper is well-written and clearly presented. The experiments were carefully conducted, and the data support the authors' conclusions. However, as currently written, I have several specific comments that prevent me from recommending this paper for publication in its current form. With a significant amount of effort, the authors may be able to address these concerns, and if that is done, the paper may be acceptable for publication.

1. In the abstract, I suggest revising the objective of the manuscript and adding more

quantitative results; be as specific and detailed as possible for the reader.

Response:

We gratefully appreciate your valuable suggestion. Quantitative results were added in the abstract, and we have added the sentence “Compared with commercial suspensions, the encapsulation by the polymer improved the deposition of TEB, and the contact angle on foliage was lowered by 40.0°. Due to small size and strong van der Waals interactions, the anti-washing efficiency of TEB NPs was increased by 37% in contrast to commercial ones. Finally, the acute toxicity of TEB NPs to zebrafish showed a more than 25-fold reduction as compared to commercial formulation indicating good biocompatibility of the nano-pesticides.” into the abstract (line 9 - line 14, page 1).

2. In the introduction, when discussing key studies from the literature, be as detailed, specific, and quantitative as possible.

Response:

Thank you for your suggestion. The discussion of key studies of pesticide droplet deposition and adhesion in the introduction has been revised to add more specific descriptions to make it clearer to the readers. More description was added on line 14 - line 21, page 2.

3. I suggest using Scheme 1 as Graphical Abstract.

Response:

Thank you for your comment. As shown in the figure below, we slightly revised Scheme 1 and put it as the graphical abstract.

4. I suggest that the TEM of Fig.1-i be performed at the same magnification as in Figure 1-h. Moreover, it is necessary to show the recrystallization process discussed by the authors.

Response:

Thank you for your valuable comment. We re-performed the TEM images of nano-pesticides obtained by TM, as shown in Fig. 1-i. As seen from the revised image, the morphology of the NPs prepared by the TM method indicates it is more like an aggregated structure rather than a recrystallized morphology. Therefore, the expression of "obvious recrystallization" was substituted by "obvious aggregation" (line 11, page 8). The TM method does not provide sufficient homogenous supersaturation distribution in the preparation of NPs, resulting in unstable NPs and aggregation over time (*Chem. Eng. J.* **2023**, 452, 139343). We thank again for the reviewer's important comment.

Figure. 1 (i). TEM images of TEB NPs via TM method, with 0.1 mg/mL TEB and 0.05 mg/mL PDMAEMA-*b*-PCL.

5. The authors describe that “the surrounding water further restricted the hydrophobic active ingredient (TEB) in the core of the NPs.” Could the TEB also be associated with the polymer wall? What type of nanocarrier was fabricated (nanosphere or nanocapsules)? I also suggest discussing this related to the active ingredient release profile information - "The similar initial release profiles at different pH values may be attributed to TEB near the NPs surfaces.”

Response:

We gratefully appreciate your valuable suggestion. For the sentence "the surrounding water further restricted the hydrophobic active ingredient (TEB) in the core of the NPs.", we tried to explain that the hydrophobic TEB was limited to diffuse away from the core of the NPs as the organic solvent was dialyzed. The TEB molecules nucleated and grew at the hydrophobic chain of the polymer during the turbulent mixing process, but at a certain size, the growth was inhibited through steric inhibition inside the polymer chains. Hydrophobic TEB was limited to diffuse out from the core of the NP as the organic solvent was dialyzed. Therefore, we believe most of the TEB was encapsulated inside the hydrophobic core. We added more comments in line 14 - line 18, page 7.

While, as the reviewer mentioned, with a high encapsulation efficiency, the TEB molecules near the core were exposed to water especially at acidic conditions as the hydrophilic chain becomes protonated and the PDMAEMA chains are more expanded in water. This is why the molecules could get a faster release at the initial

stage due to the diffusion process (*J. Mater. Chem.* **2010**, *33*, 6935-6941). We added more comments in line 3 - line 6, page 12 to make this issue less ambiguous. In addition, the particle structure was schemed clearer as shown in Scheme 1. Furthermore, a magnified particle was shown as the inset of Fig. 1-h to confirm the core-shell structure of the NPs.

Figure. 1 (h). TEM images of TEB NPs via FNP method with 0.1 mg/mL TEB and 0.05 mg/mL PDMAEMA-*b*-PCL.

Scheme 1. Preparation of TEB NPs and the application of nano-pesticides on leaves at different temperatures.

6. Rain-fastness wash-off tests: Fluorescence microscopy was used to compare the adhesion and washing resistance of TEB NPs, SC, WP, and free TEB solution. However,

what guarantees that FITC is linked to the non-encapsulated TEB? Would the rain-washing process have only eliminated FITC, not the pesticide from the leaves in Figure S5? Please further discuss it.

Response:

Thank you for your valuable comment. As reported in literatures, FITC was commonly added to SC and WP solution as a visualization method to track pesticides (*Chem. Eng. J.* **2022**, 435, 134861; *ACS Sustainable Chem. Eng.* **2020**, 8, 16555-16564). We have performed the following experiments to further support the attachment of FITC to SC and WP. Commercial SC and WP were dispersed individually in 20 mL of water, and then 0.2 mg of FITC was added into the suspensions and stirred at 500 rpm for 24 h. FITC-SC and FITC-WP (2 mL of each) were placed into a dialysis bag (MWCO, 3500 Da), which was placed in 18 mL of dialysate (ultra-pure water) with agitation at 200 rpm. After 1 hour, 3mL of the dialysate was taken out and the fluorescence intensity was determined. As shown in Supplementary Fig. 5, the fluorescence intensity of dialysate was much lower than that of commercial SC and WP diluted at the same ratio as the dialysate, which indicated that FITC was encapsulated by SC and WP. For the free TEB solution, we made an assumption that TEB and FITC will be washed away together since they have very similar hydrophobicity reflected by ACDLogP values: ACDLogP (octanol-water partition coefficient) was 4.00 ± 0.98 , and that of TEB was 3.58 ± 0.66 , which could be assumed that FITC and TEB had similar solubility. Therefore, FITC and TEB had similar adhesion energy on the leaves, and both were eliminated during the rainwater washing. We added Supplementary Fig. 5 on page 2 of supplementary information to explain this issue clearer and more description was added in line 6 - line 9 on page 16 to make this issue less ambiguous.

Supplementary Figure 5. Fluorescence spectra of (a) commercial WP and dialysate of commercial WP, and (b) commercial SC and dialysate of commercial SC.

7. The authors reported that the block copolymer PDMAEMA-*b*-PCL is excellent biocompatibility, biodegradability, and non-toxic. However, TEB-loaded PDMAEMA-*b*-PCL will have a new fate and behavior in the environment. Although the risk assessment of this nanopesticide was not studied in the manuscript (which is acceptable given the number of results presented). I would suggest that something related to the "ecotoxicological aspects" of these nanocarriers will be briefly discussed in the manuscript.

Response:

Thank you for your rigorous consideration. The analysis and discussion of potential environmental toxicity and risks for human consumption have been added and were presented in "Safety assessment of TEB NPs". Two typical toxicity evaluation models were added to evaluate the toxicity of TEB NPs to aquatic organisms. zebrafish were selected as model animals, and BEAS-2B cells (Bronchial Epithelium transformed with Ad12-SV40 2B) viability as a proxy for *in vitro* toxicity is an indicator of the safety of pesticides. As shown in Fig. 7, Supplementary Fig. 28 - 33, and Supplementary Table 2, the safety of TEB NPs to aquatic organisms were much higher than those of commercial TEB formulations. TEB NPs had a smaller effect on Zebrafish survival rate and BEAS-2B cell viability, which exhibited good biocompatibility. Method was added on page 35 - page 36. Fig. 7, Table 2, and results were added on page 24 - page 25.

Supplementary Fig. 28 - 33, and Supplementary Table 2 was added on page 11 - page 14 of supplementary information to show the biocompatibility of the nano-pesticide particles.

Figure 7. (a) The survival rate of zebrafish larvae (5 dpf) exposed to different TEB formulations with concentrations of 0.1, 0.5, 1, 5, and 10 µg/mL at 96 hr. (b) Cell viability of different TEB formulations with concentrations of 1, 5, 10, 25, and 50 µg/mL after 24 h of incubation. (* $P < 0.05$, ** $P < 0.01$, *** $P < 0.001$, vs TEB NPs group)

8. It needs to be clarified how the particle size (nm) of SC and WP was measured in Table 1.

Response:

We gratefully appreciate your valuable suggestion. The size of commercial SC and WP was measured with a NICOMP 380 ZLS with a scattering angle of 90° multiple times, similar to that of TEB NPs. The test method for particle size has been added and was presented in line 9 - line 10, page 28.

9. I also suggest clarifying how the analyses related to the length of lesions were performed in the plants.

Response:

Thank you for your suggestion. The length of lesions on the coleoptile was measured, and the curative efficacy (CE) was calculated using the following equation:

$$CE = \frac{C-L}{C} \times 100\%$$

where C (mm) is the length of lesions on the wheat coleoptile in a control, and L (mm) is the length of lesions on the wheat coleoptile treated with TEB suspension. The equation has been added and presented in line 1, page 35 to make it less confusing.

Response to Reviewer 2:

Reviewer #2: This manuscript reports the synthesis of BCP-stabilized nanocrystals of hydrophobic antifungal molecules prepared by flash nanoprecipitation (FNP). The resulting kinetically stabilized nanoparticles were used as leaf-adhesive particles that release encapsulated drug molecules to the plant leaves. The adhesive nature of the nanoparticles relies on the stimuli-responsive properties of PDMAEMA polymer blocks. The adhesive nanoparticles could enhance the efficiency of drug delivery to plant leaves, which could result in the reduction of the volume of applications of pesticides. Although this is a well-performed and comprehensive study, I am not convinced that this work brings enough new discoveries and insights to the readership of Nature Communications. I believe that this manuscript suits better to more specialized journals. I also have two technical questions.

1. Because PDMAEMA-*b*-PCL was used as a stabilizer, nanoparticles produced by the FNP method are primarily stabilized by PCL blocks. Then, how can the authors relate the increase in the release of pesticides with a collapse or expansion of PDMAEMA blocks?

Response:

We gratefully appreciate your valuable suggestion. PDMAEMA with a pKa of ~ 7.4 is sensitive to both pH and temperature and can be used as a hydrophilic block in amphiphilic polymers. As shown in Supplementary Fig. 3 (previous Supplementary Fig. 6), the particle size of TEB NPs increased from 95 nm to 160 nm as the pH values

decreased from 7.4 to 4.5. As we have mentioned in Comment 5, reviewer 1, this behavior was attributed to the protonation of the tertiary amine functional groups in the PDMAEMA component. At pH 4.5, the PDMAEMA chains are entirely protonated and highly stretched along the radial direction because of the geometrical constraint and the electrostatic repulsion between polymer chains, which cause TEB close to the surface to diffuse into the dialysate (*Chem. Commun.* **2003**, 3, 340-341). This explains well that the TEB would release faster at lower pH values. More comments were added in line 20 - line 22, page 11, and line 3 - line 6, page 12 to make this issue less ambiguous. We thank again for the reviewer's important comment.

Supplementary Figure 3. The size of TEB NPs at different temperatures and pH with 0.05 mg/mL PDMAEMA-*b*-PCL and 0.1 mg/mL TEB.

2. PDMAEMA block shows temperature- and pH-sensitive properties when the polymer is in solution. How can the authors make sure that the applied nanoparticles are in a wet or solution-like state?

Response:

Thank you for your rigorous consideration. The reviewer raised a reasonable comment. We acknowledged that TEB NPs via FNP performed better when used in warm and humid environments than in dry environments like deserts. Pesticides are applied to crop leaves by spraying, and when they are sprayed onto the leaf surface, the TEB NPs are in solution. The spraying of pesticides is performed in the morning

or when the temperature is low. After the spraying of pesticides, the NPs adhere to the leaves under wet conditions, and with the increase in temperature, TEB NP undergoes a temperature-stimulate response. On the other hand, pH-responsive properties of TEB NPs occur within the apoplastic space of the plant, which is an aqueous solution medium. We added more comments in line 17 - line 19, page 18 to make this issue less ambiguous.

Response to Reviewer 3:

Reviewer #3: The manuscript titled “Effective Deposition and Water Repelling of Temperature-Responsive Nano-pesticides on Leaves” demonstrated a nano-formulated tebuconazole prepared by flash nanoprecipitation, which had good stimuli-release properties, thermal stability, photostability, as well as better leaf adhesion, effective deposition and water repelling than commercial formulations. Although the topic is of great interest, the manuscript in its actual form is suffering from some issues that will need major revisions.

1. The measurement of particle size after two months of storage at room temperature does not fully reflect the stability of nano-formulation. The stability of the NPs should be evaluated under different temperature according to the general principles. Specifically, three parallel samples were prepared and sealed. One sample was stored at 0 ± 2 °C for 7 d, the other two samples were stored separately at 25 ± 2 °C and 54 ± 2 °C for 14 d, and the particle size and PDI of the samples were measured. The author may refer to the methods in the following literature: “Synthesis and characterization of a novel stimuli-responsive zein nano delivery system for the controlled release of emamectin benzoate, Want et al., 2022, Environmental Science Nano, 9, 4411-4422.”

Response:

We gratefully appreciate your valuable suggestion. The stability of size and PDI of TEB NPs at 0 °C, 25 °C, 38°C, and 54 °C were performed. As shown in Supplementary

Fig. 1, the average size and PDI of TEB NPs remained essentially little changed at 0 °C, and 38 °C during storing for 3 weeks, while the size of TEB NPs at 54 °C was increased from 75 nm to 200 nm, PDI was increased from 0.16 to 0.3, as the LCST of the polymer was ~ 40 °C, beyond which the particles gradually aggregated. The results have been added on line 22, page 7, line 1 - line 3, page 8, and were presented as Supplementary Fig. 1 on page 1 of supplementary information.

Supplementary Figure 1. Stability of (a) size and (b) PDI of TEB NPs at 0 °C, 25 °C, 38 °C, and 54 °C.

2. Although the author mentioned that nanocarriers are non-toxic, since this nano-formulation can increase the deposition of pesticides on crop leaves and change the transfer of pesticide within crops, the author should consider whether pesticide residues will increase in the edible parts of the crops.

Response:

Thank you for your rigorous consideration. To study the nano-pesticide residues in the edible parts of the crops, tomato plants were selected, and a middle leaf was treated with 1 mL of FITC-loaded TEB NPs. At 24 hours after treatment, the untreated section of the leaf and the tomato flesh portion adjacent to the treated position were sampled for imaging observation with CLSM. As shown in Supplementary Fig. 9, the fluorescence intensity of the tomato flesh portion was lower than that of the leaf, which indicated that pesticide residues did not increase significantly in the edible parts of the crops. More comments were added in line 14 - line 16, page 20, line 1 - line 4, page 33, and Supplementary Fig. 9 was added on

Supplementary Figure S9. CLSM images of (1-3) the untreated section of the leaf adjacent to the treated position, and (4-6) the untreated section of the tomato flesh portion adjacent to the treated position.

3. The author conducted statistical analysis of the experimental data, but in the methods section, no information on the software used for statistical analysis was provided.

Response:

Thank you for your suggestion. The statistical analysis has been added and was presented in “Statistical analysis” (line 6 - line 12, page 36).

4. The description of some results is incorrect. For example, in the second paragraph of section 2.2.1 Stimuli response release of TEB NPs, “The cumulative release obtained at 50 °C is 94%, higher than 70% at 30 °C and 43% at 40 °C after 48 h.”, which is not consistent with Figure 2 (a).

Response:

Thank you for your suggestion. We are very sorry for the errors. We have changed

the content to "The cumulative release obtained at 50 °C is 94.5%, higher than 73.6% at 30 °C and 80.2% at 40 °C after 48 h" (line 8, page 11). We have thoroughly checked and corrected the errors in our revised manuscript. Thank you again for your suggestion.

REVIEWERS' COMMENTS

Reviewer #1 (Remarks to the Author):

The authors have sufficiently addressed my technical and novelty concerns for the article through their revisions, and I find this work to be publishable in Nature Communications in its current form.

Reviewer #2 (Remarks to the Author):

The authors revised the manuscript thoroughly by addressing reviewers' comments. I believe this version of the manuscript is significantly improved and adequate for publication.

Reviewer #3 (Remarks to the Author):

All my concerns have been addressed in the revisions.
In terms of formulation innovation, the author's work has basically covered all aspects of the characterization and evaluation of nano-pesticide formulations.